neuroscience

neural plasticity, calcarine cortex, blindness

**Author for correspondence:**
Lore Thaler
e-mail: lore.thaler@durham.ac.uk

# Retinotopic-like maps of spatial sound in primary 'visual' cortex of blind human echolocators

Liam J. Norman and Lore Thaler

Department of Psychology, Durham University, Durham DH1 3LE, UK

LJN, 0000-0002-6119-2646; LT, 0000-0001-6267-3129

The functional specializations of cortical sensory areas were traditionally viewed as being tied to specific modalities. A radically different emerging view is that the brain is organized by task rather than sensory modality, but it has not yet been shown that this applies to primary sensory cortices. Here, we report such evidence by showing that primary 'visual' cortex can be adapted to map spatial locations of sound in blind humans who regularly perceive space through sound echoes. Specifically, we objectively quantify the similarity between measured stimulus maps for sound eccentricity and predicted stimulus maps for visual eccentricity in primary 'visual' cortex (using a probabilistic atlas based on cortical anatomy) to find that stimulus maps for sound in expert echolocators are directly comparable to those for vision in sighted people. Furthermore, the degree of this similarity is positively related with echolocation ability. We also rule out explanations based on top-down modulation of brain activity—e.g. through imagery. This result is clear evidence that task-specific organization can extend even to primary sensory cortices, and in this way is pivotal in our reinterpretation of the functional organization of the human brain.

## 1. Introduction

Building an accurate model of sensory processing in the human brain provides the foundation for successful sensory restoration and substitution [1]. There is mounting evidence to suggest that cortical processing areas are better understood in terms of their task-specificity, not sensory-specificity [2–5], but so far this evidence is constrained to higher-order sensory areas. By contrast, such evidence is still missing for the primary sensory areas—that is, those areas that serve as the first cortical site of sensory processing. Specifically, while it has been shown that primary sensory areas may process input from atypical modalities [6–13], it remains to be shown that these areas can use this input in order to perform a specific task that would ordinarily be performed with input from its typical modality.

In order to test whether task-specificity extends to primary sensory areas, it must be shown that the characteristic topographical maps of sensory input within these areas—e.g. the retinotopic map of visual space in primary 'visual' cortex [14] or the tonotopic map of sound frequencies in primary auditory cortex [15]—serve a directly analogous purpose for input provided by a different modality [16]. There is evidence based on patterns of non-stimulus-driven activity (i.e. resting state data) that the intrinsic retinotopic organization of early visual cortex can be preserved even in congenitally blind individuals [17]. This means that neurons within early visual cortex are connected in a way that is consistent with a retinotopic organization. It is unknown, however, whether this intrinsic connectivity can be adapted by a non-visual modality for the mapping of space. There is also one previous study that used transcranial magnetic stimulation (TMS) to map the topographical representation of somatosensory input in

visual cortex [18], but the cortical sites stimulated in that study covered many visual association areas in addition to primary 'visual' cortex. The presence of a cross-modal topographical stimulus map within primary 'visual' cortex, therefore, remains to be shown.

Here, we test if primary 'visual' cortex in blind humans who are proficient in echolocation can be used to map the spatial layout of sounds in a manner that is directly analogous to the retinotopic mapping of visual input. Echolocation is the ability to perceive space through sound echoes [19], and some people who are blind use mouth-click-based echolocation on a regular basis to perceive the space around them. Here, we used human echolocation as a model to study cross-modal processing of space in primary 'visual' cortex because early visual cortex in expert echolocators is known to be engaged by echo processing [7] and, furthermore, higher task-specific 'visual' brain areas are engaged by echolocation—e.g. for shape perception [20] or material perception [21]. Finally, it has also been shown that expert human echolocators are capable of resolving very fine spatial positions of objects through echoes [22].

Blind expert echolocators, blind controls, and sighted controls took part in the study. We used sparse-sampling functional magnetic resonance imaging (fMRI) [23] to map neural responses to sensory stimulation at eight horizontal spatial eccentricities (echo sounds and source sounds in all participants, and also vision in sighted participants). We predicted that, for individuals who are blind and experts in echolocation, there will be evidence of retinotopic-like mapping of echo sounds in primary 'visual' cortex (i.e. there should be contralateral mapping of stimuli, and stimuli at greater eccentricities should be mapped at more anterior points), but not for blind or sighted controls. We also tested if this retinotopic-like mapping of spatial sounds is echo-specific, or whether it generalizes to other acoustic stimuli that do not require echolocation expertise in order to be perceptually resolved (i.e. source sounds). Visual stimulation in sighted controls was included to obtain baseline comparison measurements.

## 2. Results

### (a) Quantifying retinotopic-like mapping in primary 'visual' cortex

Phase-encoded eccentricity mapping of echo sounds, source sounds, and visual stimuli (in sighted controls) was carried out individually for each participant. This was done using a voxel-based cross-correlation analysis, where the functional data time-series was correlated with eight shifted (lagged) copies of a regressor describing the time-course of stimulus presentation at one spatial location (this regressor consisted of 'ones' where the stimulus was present, and 'zeros' where absent). Based on this cross correlation, each voxel was labelled with a value to denote the stimulus position with which it most strongly correlated (i.e. −40, −20, −10, −05, +05, +10, +20, or +40, where negative values denote left space and positive ones right space). Phase-encoded maps of echo and source sounds for expert echolocators are shown in figure 1, and for blind controls in figure 2. Phase-encoded maps for sighted controls are shown in electronic supplementary material, figures S1–S3. All maps are displayed on inflated cortical surface views of primary 'visual' cortex. It is important to note that neural data are the result of a cross-correlation analysis, and as such they do not represent the strength of activity in response to auditory stimulation *per se*, but rather how consistently a specific voxel responded to a specific stimulus position.

Due to our use of a sparse-sampling scanning protocol, the phase-encoded maps that we obtained are not visually comparable to those typically reported following the use of a standard scanning protocol. See also the maps for visual stimuli that we acquired in our sighted participants using a sparse-sampling protocol for further comparison (electronic supplementary material, figure S1). It is also possible to notice some variation in the mapping of sound stimuli in the expert echolocators (EE)—for example, there is evidence of both eccentricity and contralaterality mapping in EE3 and EE5, but less so in EE1, and no obvious pattern of any kind in EE2 and EE4. Thus, in order to provide an objective and quantitative measure of the degree to which the observed maps in primary 'visual' cortex follow retinotopic-like organization, each map was compared to a probabilistic retinotopic atlas for eccentricity that was first fitted to each participant's primary 'visual' cortex [24]. The atlas has a very low prediction error for eccentricity (0.51° of visual angle), in that it accurately predicts the location and layout of a retinotopic map that is acquired through functional measurement [24]. The atlas identifies the estimated anatomical location of the stria of Gennari (anatomical marker of primary 'visual' cortex) with reference to cortical surface topology [24]. It then assigns a numeric value to each voxel in this area to represent the location in space (in visual degrees) that the voxel is most likely to represent, ranging from −90 (most peripheral left space) to +90 (most peripheral right space). We calculated Pearson's $r$ to quantify the correlation between these predicted eccentricity values and those observed in each of our experimental conditions. We use this method to quantify retinotopic-like organization because it provides a simple test of whether the pattern of neural mapping is random (i.e. a correlation coefficient of zero) or whether it is comparable to a retinotopic organization, where higher coefficient values indicate greater retinotopic-like organization for the mapping of stimuli. All voxels labelled as primary 'visual' cortex in the probabilistic atlas were entered into the correlation. These correlation coefficients, along with 95% confidence limits (derived from a bootstrapping method), are shown in electronic supplementary material, table S1. We also confirmed that correlation coefficients expected by chance are indeed zero (results shown in electronic supplementary material, table S2). We also measured the overall contralateral mapping—i.e. without taking into account eccentricity—of echo and source sounds in primary 'visual' cortex as a coarser index of functional activity consistent with retinotopic organizational principles (see electronic supplementary materials).

### (b) Quantifying echo and source sound localization ability behaviourally

We measured the ability of each participant to localize the echo and source sound stimuli in a psychophysical task after running the fMRI component (see electronic supplementary materials text and table S5). As expected, EEs had better echo localization ability compared to controls, and all participants were very good at localizing source sounds.

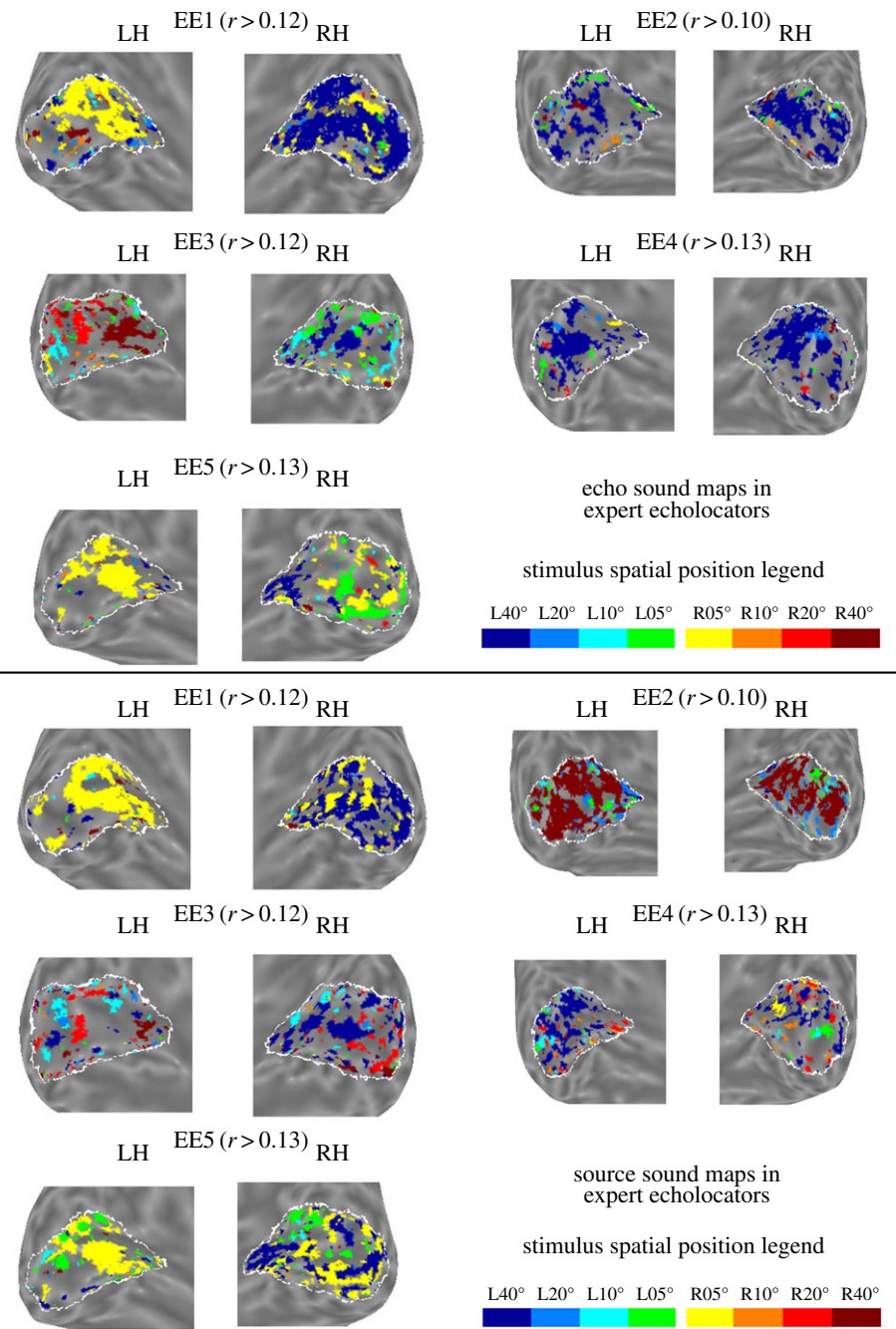

**Figure 1.** Phase-encoded stimulus maps for echo sounds (upper images) and source sounds (lower images) in primary 'visual' cortex of blind expert echolocators. Each pair of images shows inflated cortical surface views of left and right primary 'visual' cortex for each participant (EE1 to EE5). Each voxel in primary 'visual' cortex is colour-coded to indicate the stimulus position with which the voxel's activity was correlated most strongly. The colour-coded legend for spatial position is shown in the bottom right. To aid with visualization, only voxels above a correlation coherence threshold are colour-coded (this value is shown beside each participant identifier). The coherence threshold is set individually for each participant but is fixed across stimulus conditions (echo/source/vision (for sighted participants)). All voxels in primary 'visual' cortex were included in the statistical analyses reported in the results section. The white outline on each image shows the boundary of primary 'visual' cortex as defined by the probabilistic atlas used [24]. (Online version in colour.)

## (c) Is there retinotopic-like mapping of echo sounds in primary 'visual' cortex of expert echolocators?

Figure 3a shows the degree of retinotopic-like mapping of echo sounds plotted against echo localization ability for all participants. In order to determine whether the degree of retinotopic-like mapping of echo sounds is predicted by echolocation expertise and/or blindness alone, and also echo localization ability, a multiple linear regression analysis was carried out with the predictors 'Echolocation Expertise', 'Blindness', 'Echo Localization Ability', and the interaction between 'Echolocation Expertise' and 'Echo Localization Ability'. 'Echolocation expertise' was a categorical predictor to compare expert echolocators to the remaining participants, and 'Blindness' was a categorical predictor to compare all blind participants to sighted participants. We used the method of multiple linear regression here because it allows us to include all relevant factors in a single analysis. The regression revealed a significant effect of 'Echolocation Expertise' (standardized beta = −3.893; $t_{10}$ = 3.675; $p = 0.004$; unstandardized $B = −1.015$, 95% CI = [−1.630, −0.400])[1] and a significant interaction effect (standardized beta = 4.341; $t_{10} = 4.341$; $p = 0.001$; unstandardized $B = 1.472$, 95% CI = [0.716, 2.227]). None of the other

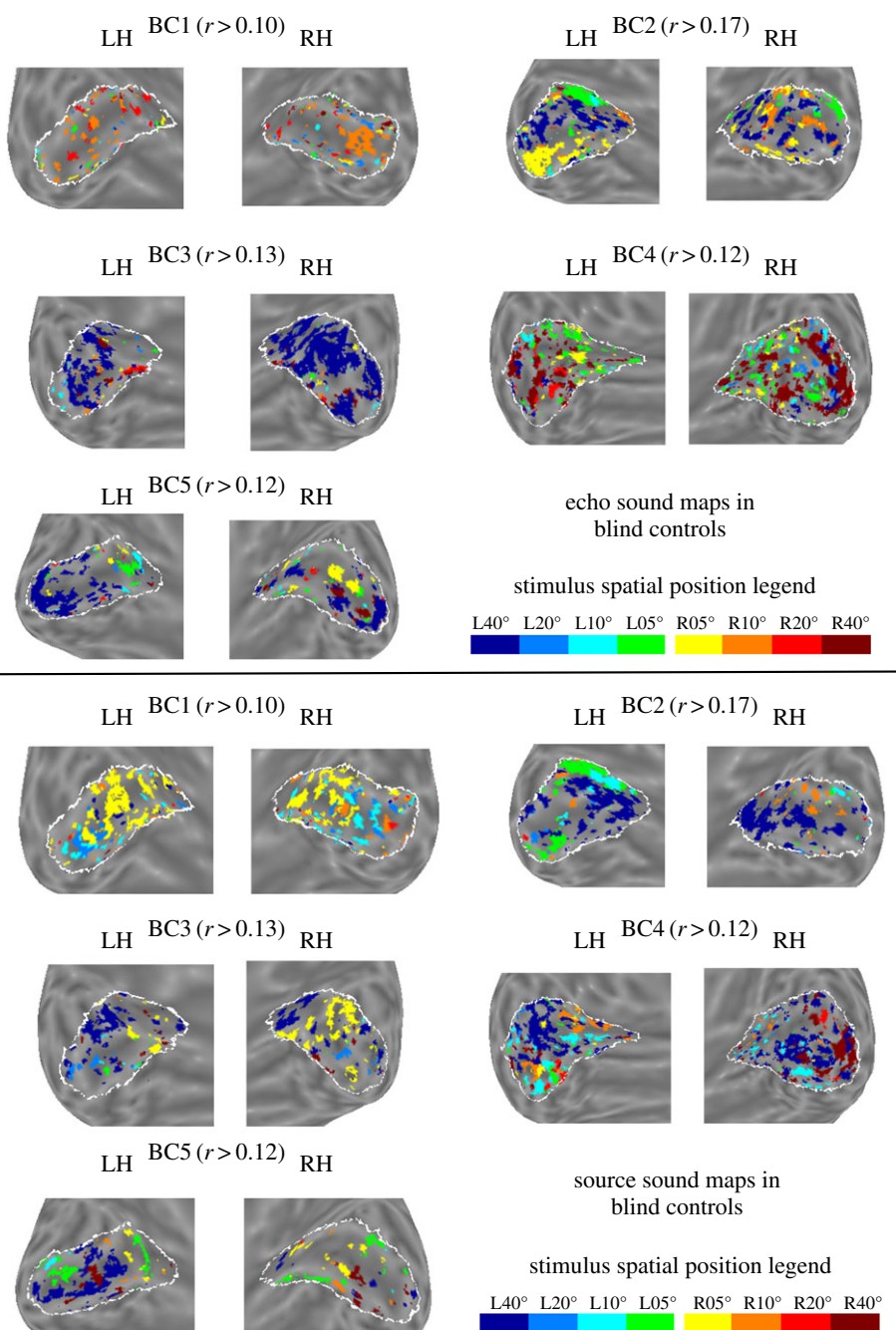

**Figure 2.** Phase-encoded stimulus maps for echo sounds (upper images) and source sounds (lower images) in primary 'visual' cortex of blind controls. Format for figure 2 follows the format for figure 1. See figure 1 legend for details. (Online version in colour.)

predictors were significant. Follow-up individual correlation analyses between the measures 'Echo Localization Ability' and the degree of retinotopic-like mapping for experts and non-experts showed a significant positive correlation for echolocation experts ($r_3 = 0.963$; $p = 0.008$), but not the other participants ($r_8 = -0.307$; $p = 0.388$). The overall model explained 80.9% of the variance ($R^2$) in retinotopic mapping of echo sounds, which was significant ($F_{4,14} = 10.559$, $p = 0.001$). This indicates that, for expert echolocators with high echolocation ability, primary 'visual' cortex was more likely to map the spatial locations of echoes in a retinotopic-like manner. The same pattern of results was found with respect to the degree of contralateral mapping of echo sounds (i.e. regardless of eccentricity). This is described in the electronic supplementary materials and shown in figure S4a.

Separate single-case statistics [26,27] confirm that EEs 1, 3, and 5 have a significantly higher degree of retinotopic-like mapping of echoes compared to controls (see electronic supplementary materials). Further analyses showed that this mapping in EE3 and EE5 can be explained by mapping of both laterality and eccentricity, while for EE1 the effect is driven by laterality (see electronic supplementary materials).

### (d) Is there retinotopic-like mapping of source sounds in primary 'visual' cortex?

Figure 3b shows the degree of retinotopic-like mapping of source sounds plotted against echo localization ability for all participants. A multiple linear regression analysis was carried out using the same predictors as those used in the previous analysis[2] but with the degree of retinotopic-like

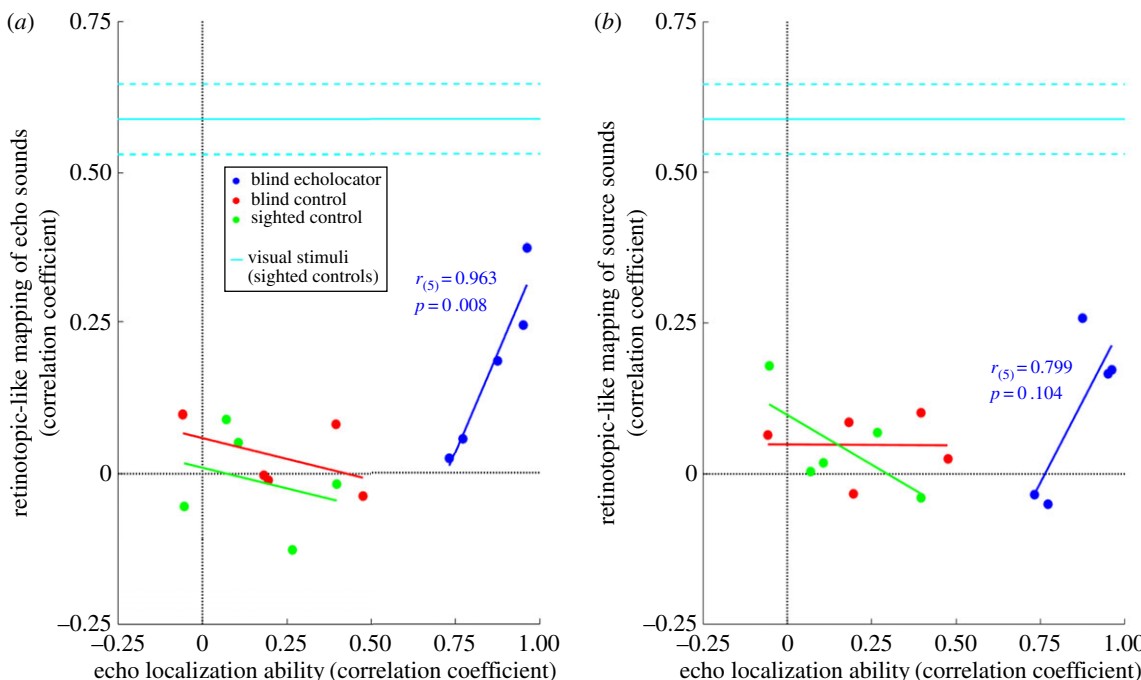

**Figure 3.** (*a*) The association between the degree of retinotopic-like mapping of echo sounds in primary 'visual' cortex (*y*-axis), echolocation expertise (separate colours), and echo localization ability (*x*-axis). The degree of retinotopic-like mapping of echo sounds in primary 'visual' cortex is significantly associated with echolocation expertise and the interaction between expertise and echo localization ability. For the expert echolocators, the degree of retinotopic-like mapping of echo sounds is significantly associated with their echo localization ability. The solid horizontal line in cyan shows the mean correlation coefficient between the observed eccentricity maps for visual stimuli in sighted participants (described in the electronic supplementary materials) and the predicted retinotopic atlas (dotted lines show ±1 s.e. of the mean), and therefore represents a practical upper limit for the extent of retinotopic-like mapping of echo/source sounds. (*b*) The same as in (*a*) but for the mapping of source sounds. Just like for echo sounds, the degree of retinotopic-like mapping of source sounds in primary 'visual' cortex was significantly associated with echolocation expertise and the interaction between expertise and echo localization ability. Yet, the overall model (and the individual correlation coefficient) was not significant, suggesting that associations are weaker as compared to those for echo sounds in our study.

mapping of source sounds as the dependent variable. The regression revealed a significant effect of 'Echolocation Expertise' (standardized beta = −4.521; $t_{10} = 2.675$; $p = 0.023$; unstandardized $B = −0.878$, 95% CI = [−1.610–0.147])[3] and a significant interaction effect (standardized beta = 5.297; $t_{10} = 2.956$; $p = 0.014$; unstandardized $B = 1.191$, 95% CI = [0.293 2.089]). None of the other predictors was significant. Follow-up individual correlation analyses between 'Echo Localization Ability' and the degree of retinotopic-like mapping separately for experts and non-experts showed non-significant correlations for echolocation experts ($r_3 = 0.799$; $p = 0.104$), as well as the other participants ($r_8 = −0.368$; $p = 0.295$). The overall model explained 51.2% of the variance ($R^2$) in retinotopic-like mapping of source sounds, which was non-significant ($F_{4,10} = 2.628$; $p = 0.098$). This suggests that, while the degree of retinotopic-like mapping of source sounds in primary 'visual' cortex is significantly associated with echolocation expertise and the interaction between expertise and echo localization ability, the explanatory power of these factors is weaker than those for the mapping of echo sounds. The same pattern of results was found with respect to the degree of contralateral mapping of source sounds (i.e. regardless of eccentricity). This is described in the electronic supplementary materials and shown figure S4b.

Separate single-case statistics [26,27] confirm that EE5 has a significantly higher degree of retinotopic-like mapping of source sounds compared to controls (see electronic supplementary materials). Further analyses showed that

this mapping in EE5 can be explained by mapping of both laterality and eccentricity (see electronic supplementary materials).

While for theoretical reasons the main focus of our manuscript is on V1, we also carried out the same mapping analyses in the second and third visual cortex (V2 and V3), each of which contains a retinotopically organized map of visual space [28] and is directly connected to auditory cortex [29–31]. We also measured similarity of maps in V2 and V3 to those observed in V1. For those EEs where we found retinotopic-like mapping of echo and source sounds in primary 'visual' cortex, we also find it in V2 and V3, suggesting that retinotopic-like mapping of echo and source sounds extends beyond the primary 'visual' area. Furthermore, high similarity between maps in V1 and V2/V3 strongly suggest that maps in primary 'visual' cortex are reliable and not the result of low statistical power. All of this is reported in the electronic supplementary materials.

## 3. Discussion

In this study, we found evidence for the retinotopic-like mapping of sound echoes in blind individuals who are experts at perceiving space through sound echoes using clicks, in comparison to blind and sighted controls. Importantly, the degree of retinotopic-like mapping of sound echoes was positively associated with echo localization ability. Those individuals who showed retinotopic-like mapping of sound echoes also

showed evidence of spatial mapping of source sounds, even though statistical explanatory power of regression models applied to those data was weaker. This weaker mapping for source sounds, as compared to echo sounds, might be explained by the fact that the typical brain areas that are responsible for spatially mapping source sounds are still intact in all our participants, and so there is perhaps less need for the brain to undergo neural reorganization in order to map these in V1. Nonetheless, taken together our data support the conclusion that the characteristic functional topography of a primary sensory area—here, the retinotopic map in primary 'visual' cortex—can be used to map sensory input from an atypical modality for a directly analogous task-specific purpose—here, localization of sound.

The lack of retinotopic-like mapping of source sounds (as well as echo sounds) in our control participants demonstrates that this mapping in experts does not simply arise through top-down modulation of neural activity (e.g. through mental imagery [32]). If this were true, then there should be clear evidence of retinotopic-like mapping of source sounds in all participants because all participants had a very high ability to resolve the source sound positions (see electronic supplementary material, table S5). Alternatively, if one were to argue that the results could be explained by a combination of neuroplastic changes due to blindness and imagery, then we would expect at least source sounds to map in our blind control participants, but there was no evidence for this.

The results of this study are the first evidence that activity even in early sensory areas can be flexibly organized by task, and not necessarily by sensory modality. The afferent projections of primary sensory areas mostly consist of those from subcortical structures that relay information directly from the external sense organs, and so their functional role has classically been considered to be more strictly constrained to a specific modality. Direct evidence for the task-specific theory of brain organization has so far been limited to higher-order sensory areas [16,33–35], but here we have shown that the characteristic functional topography of a primary sensory area can be used for a sensory-independent common purpose. This evidence is pivotal in our reinterpretation of the functional organization of sensory cortex, and follows previous important results that, while highlighting the existence of intrinsic retinotopic connectivity [17] or cross-modal division of labour in primary 'visual' cortex [6], did not show that retinotopic maps in primary 'visual' cortex can be adapted by a non-visual modality in a task-specific fashion.

We observed a correlation between echolocation ability and the degree of retinotopic-like mapping of sound echoes in expert echolocators, which implies a functional link between behavioural ability and the degree of neural reorganization. This link between behavioural ability and neural reorganization is consistent with previous work [12], which showed a strong link between monaural sound localization ability and the strength of its related neural activity in occipital cortex of blind individuals. Given that previous studies have shown that functionally reorganized visual areas are in fact causally involved in processing non-visual information—e.g. through the application of neurostimulation [36,37]—it is possible that primary 'visual' cortex is functionally necessary for the perception of space through sound, and in particular sound echoes, in some expert echolocators.

What mechanisms might underlie the spatial mapping of sounds in primary 'visual' cortex? We propose that, because the spatial pattern of this mapping is directly comparable to that of visual space in the sighted brain (i.e. it follows a retinotopic-like pattern), it is likely that this mapping takes advantage of the intrinsic retinotopic organization of early visual cortex, which forms even in the complete absence of visual input [17]. An analogous neural map of space does not exist in primary auditory cortex (it is tonotopic [38]), and so the map of space in primary 'visual' cortex might be the most suitable cortical site on which to map spatial location as conveyed through sound. One explanation for cross-modal reorganization of 'visual' cortex is that primary 'visual' cortex is more strongly connected to other sensory areas, including auditory cortex [39], in cases of early or prolonged vision loss. These greater structural connections might confer additional functional connectivity between auditory and visual areas when an individual performs a relevant auditory task, but not necessarily at rest [40]. It is clear from the results of the present study, however, that blindness is not sufficient to explain the cross-modal recruitment of primary 'visual' cortex, as blind control participants did not show evidence of retinotopic-like mapping either for source or echo sounds. It is important to note, however, that there was variation in the onset of blindness in our sample, i.e. BC5 is classified as late blind and EE3, BC2, and BC4 (while early blind) have no clear onset of total blindness, and that also the age at which EEs started using echolocation is variable. However, there was no relationship between degree of retinotopic-like mapping and age at onset of blindness in either group, and in our EE sample there was also no relationship between neural mapping and age at onset of echolocation use. Furthermore, because the degree of retinotopic-like mapping of sound in the expert echolocators was positively associated with echo localization ability, proficiency in processing sensory input might be the critical factor in the neural reorganization of primary sensory areas. This is in line with previous studies showing cross-modal activity in early 'visual' cortex from proficient sensory skill use (e.g. Braille, or sensory substitution devices [10]; monaural sound localization [12]).

The results of this study have implications for successful visual rehabilitation through sensory prostheses (e.g. retinal or cortical implants). Such methods have thus far produced limited results in cases of early blindness [10], with one significant obstacle being the arduous rehabilitative period that follows the restoration of visual input. In such cases, it is likely that early visual cortex has undergone functional reorganization to process non-visual input in a manner that does not directly map onto its processing of visual input (e.g. [6]). The results from the current study leave open the possibility that for individuals who have a stronger degree of intact functional topography in primary 'visual' cortex, rehabilitation would be less arduous if normal visual input were restored, as the input could be mapped onto a pre-existing retinotopically organized sensory map.

In conclusion, we have shown clear evidence that task-specific sensory-independent organization of the human brain extends even to a primary sensory area. Although it is inarguable that primary sensory areas preferentially process input from one modality over others, they nonetheless retain the ability to carry out at least some of the characteristic tasks when relevant information is provided through another

sensory modality. This is pivotal in our interpretation of the functional organization of the human brain.

# 4. Material and methods

## (a) Participants

A total of 15 male participants took part in the experiment: five blind expert echolocators, five blind controls, and five sighted controls (see electronic supplementary material, table S6 for details of participants' age and, where relevant, causes and degree of vision loss and history of echolocation use). Those who were classed as expert echolocators reported using click-based echolocation on a daily basis for 20 years or longer. Blind and sighted controls reported having no prior experience with click-based echolocation. Participants had normal hearing, with the exception of EE5 who has some loss for frequencies beyond 4 kHz consistent with his age, and BC3, who reported to have tinnitus. Sighted participants had normal or corrected to normal vision. All participants took part in echo and source sound mapping conditions. The sighted participants took part in an additional vision mapping condition. Participants were compensated with £10/hour for their participation.

## (b) Echo and source sound stimuli

Auditory stimuli were either echo or source sounds, each composed of binaural sound recordings. Binaural recordings were made individually for each participant (see electronic supplementary materials for further details on the recording process). For each set of echo sound recordings, a loudspeaker was placed at the participants mouth facing straight ahead, and a sound reflector (wooden disc, 17.5 cm diameter) was placed directly facing the participant at ear height (mounted on a pole, 1 cm diameter) at a radial distance of 1 m and at one of eight angular (azimuth) positions: L40°, L20°, L10°, L05°, R05°, R10°, R20°, and R40°, where L and R correspond to left and right relative to the participant's orientation. By contrast, for each set of source sound recordings, the loudspeaker was positioned at one of the eight angular positions, directly facing the participant. The same artificial click emission used in the echo sound recordings was used for the source sound recordings, but here it is emitted directly towards the participant and with no other sound reflectors present. Several recordings of individual clicks were made in order for us to exclude those that contained undesired sounds (e.g. caused by participant breathing, experimenter movement, etc.). Illustrations of the apparatus and set-up for echo and source sound recordings are shown in figure 4a,b,d, and examples of recorded waveforms of echo and source sound stimuli are shown in figure 5. Audacity (2.1.2, 2016) was used for inspection and digital cutting of the sound recordings to create digital wav-files for each of the eight stimulus locations. The duration of each of these files was 8 s long and contained 10 equally spaced clicks (a rate of 1.25 clicks/s). Acoustic analysis of these sound stimuli, along with details on equipment used for recording and playback, are included in the electronic supplementary materials.

## (c) MRI scanning procedures

Imaging was performed on a 3-Tesla, whole-body MRI system (Magnetom Tim Trio; Siemens, Erlangen, Germany) and 32-channel head coil. A T1-weighted, optimized sequence (MP RAGE) was used to acquire $1 \times 1 \times 1$ mm resolution structural images for each participant. A single-shot gradient echo-planar pulse sequence in combination with a sparse-sampling design [23] was used to acquire $3 \times 3 \times 3$ mm resolution functional images. Participants wore MRI-compatible insert earphones

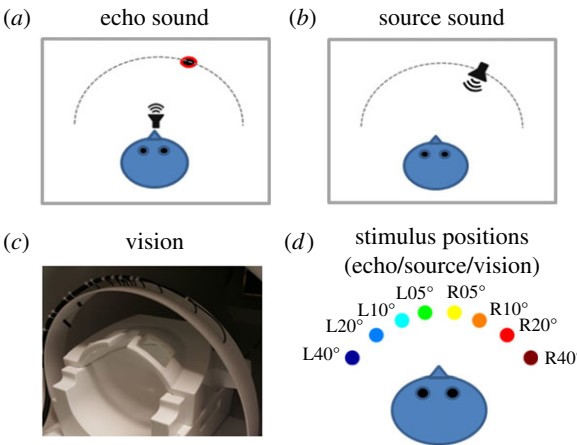

**Figure 4.** (a) Echo sound stimuli were recorded with inner-ear microphones as a loudspeaker in front of the participant emitted an artificial mouth click. An object (wooden disc, 17.5 cm in diameter) at a distance of 1 m, and at varying eccentricities, reflected the sound that was emitted by the loudspeaker back to the participant. Recordings were made inside an anechoic room. These recordings were played to participants inside the MRI scanner using MRI-compatible insert earphones. (b) Source sound stimuli were recorded under the same conditions, but with the loudspeaker directly facing the participant at a distance of 1 m, and at varying eccentricities and with no other sound reflectors present. (c) Visual stimuli were presented to participants (sighted only) inside the MRI scanner using a custom-built apparatus. End-emitting fibre optic filaments were used to provide red light stimuli to participants at different eccentricities as they lay inside of the MRI scanner. Details of the apparatus can be found in the electronic supplementary materials. (d) The same eight stimulus eccentricities were used (L40°, L20°, L10°, L05°, R05°, R10°, R20°, R40°; relative to the participant's central straight-ahead direction (and in vision conditions: central fixation)) across the three experimental conditions.

(model S-14, Sensimetrics, Malden, MA) encased in noise attenuating foam tips for all functional runs. For more details, see electronic supplementary materials.

All participants took part in the echo and source sound conditions. Sighted participants additionally took part in the vision condition. Separate functional runs were carried out for echo sound, source sound, and vision conditions. For each experimental condition, eight functional runs were carried out (except for participant EE5, where only seven runs were carried out for both the echo and source sound conditions). Each run contained 50 functional volume acquisitions—the first two corresponded to baseline trials, in which no stimulus was presented to the participant, and the remaining 48 corresponded to stimulus trials. In each of the 48 stimulus trials, the stimulus at a single location was presented for a period of 8 s followed by a tone (50 ms, 1200 Hz). A single functional volume (lasting 2 s) was acquired following the 8 s stimulus presentation. Across the 48 stimulus trials, the eight stimulus locations were presented to participants in a fixed ordinal sequence from left to right. This allowed participants in all conditions (i.e. even non-echolocating participants in the echo sound condition) to know which response button to press after each sound played. Importantly, the order was the same across all conditions for all participants. This sequence repeated six times in each run. Thus, across the eight runs there were 48 functional volumes acquired for each stimulus location. Each session took approximately 2 h to complete. Participants completed sessions on separate days.

On each trial participants pressed one of eight response keys when they heard the tone, using MRI-compatible keypads (one held in each hand) with buttons located under each of the eight fingers (thumb presses were not used). For the L40°

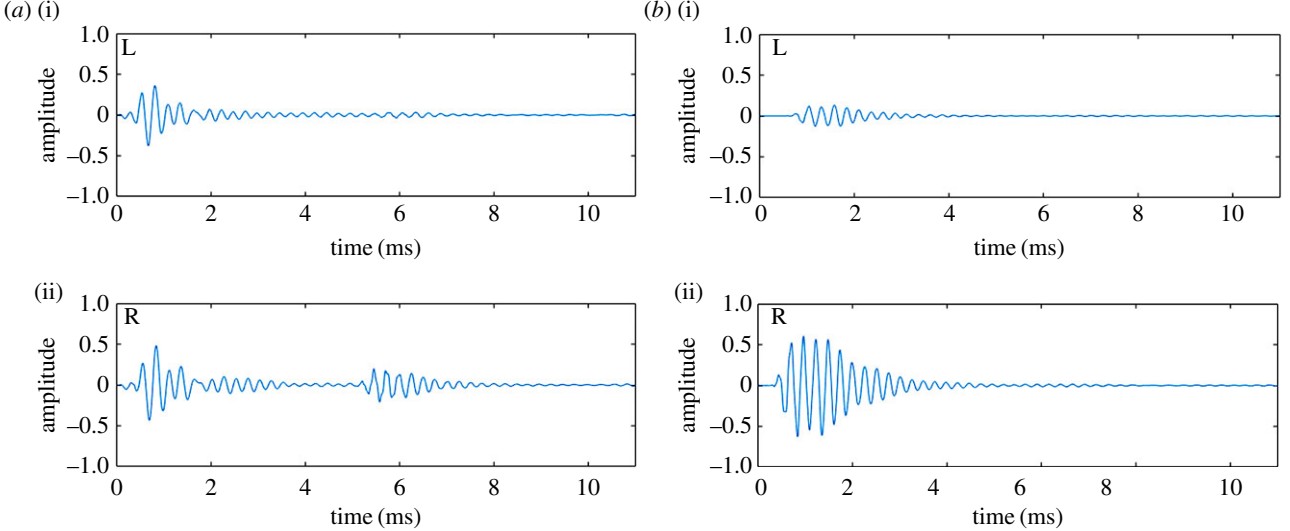

**Figure 5.** (*a*) Example waveform of a binaural echo sound recording. Panel (i) shows the left channel, and (ii) shows the right channel. The first, larger waveform represents the artificial mouth click emitted by the loudspeaker, and the second smaller waveform the reflected sound echo. In this example, the reflecting object (wooden disc) was located 40° to the right, and as a consequence the echo is more pronounced in the right ear. (*b*) Example waveform of a binaural source sound recording. Panel (i) shows the left channel, and (ii) shows the right channel. The waveform represents the click. In this example, the loudspeaker emitting the click was located 40° to the right, and as a consequence the click is more pronounced (and also earlier) in the right ear. There is no echo, because for source sounds there were no other sound reflectors present. (Online version in colour.)

stimulus location, participants would press with their left little finger; for the L20° location they would press with their left ring finger and so on.

## (d) Analysis of fMRI data

All statistical analyses were carried out individually for each participant on functional imaging data in native participant space. This is because previous studies have shown high levels of inter-individual differences in the cross-modal recruitment of primary sensory areas [41], and normalization of functional imaging data to a standard brain space introduces the potential problem of spatially blurring primary 'visual' cortex with its neighbouring areas. Pre-processing of fMRI data was carried out using FMRIB (Oxford University Centre for Functional MRI of the Brain) Software Library (FSL 5.0.10). Cortical reconstruction was carried out with the FreeSurfer image analysis suite. Data were brain-extracted (using BET [42]) and within-participant registration of low-resolution functional images to high-resolution structural (T1) images was achieved using FLIRT (7 d.f.; [43,44]. The first two functional volumes (acquired during silence trials) within each run were discarded. The following pre-statistic processing was applied to each run of functional data: motion correction using MCFLIRT [43], slice-timing correction using Fourier-space time-series phase-shifting, grand-mean intensity normalization and high-pass temporal filtering (Gaussian-weighted least-squares straight line fitting, with sigma = 100 s). The pre-processed functional data from all eight runs for each experimental condition were then concatenated in time, giving a four-dimensional dataset with 384 time points (48 trials × 8 runs), or 336 time points for participant EE5. These data were then exported to MATLAB R2015b (The Mathworks, Natick, MA), where statistical processing and data visualization was carried out using a combination of the open source software suite 'mrTools' (Gardner Lab, Stanford University, USA) and custom Matlab functions.

Ethics. Testing procedures were approved by the ethics board at Durham University, and participants gave written, informed consent prior to testing. For those participants who were blind, the consent form was read to them and the location to sign was indicated with a tactile aid. All experimental procedures conformed to The Code of Ethics of the World Medical Association as stated in the Declaration of Helsinki (1964).

Data accessibility. Data files and analysis scripts/functions are available from the Dryad Digital Repository: https://doi.org/10.5061/dryad.g614mb0 [45]. Raw imaging data relating to our participants are stored on computers in Durham University. Our participants did not give consent for these raw data to be distributed. Sound files that were used as stimuli can be obtained from the corresponding author (L.T.).

Authors' contributions. L.N. and L.T. contributed to all aspects of the manuscript.

Competing interests. The authors do not have any competing interests to declare.

Funding. This study supported by BBSRC (grant no. BB/M007847/1) to L.T.

## Endnotes

[1]The negative beta weight for 'Echolocation Expertise' in this model arises because of the presence of an interaction term [25].
[2]We use Echo Localization Ability as a predictor here, not Source Localization Ability, because our hypothesis concerns whether echolocation ability predicts the degree of mapping of echo and/or source sounds.
[3]See endnote 1.

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
