## [Reviewer comments · Proceedings of the Royal Society B: Biological Sciences]

Review History

RSPB-2019-1333.R0 (Original submission)

Review form: Reviewer 1 (Ella Striem-Amit)

Recommendation

Major revision is needed (please make suggestions in comments)

Scientific importance: Is the manuscript an original and important contribution to its field?

Excellent

General interest: Is the paper of sufficient general interest?

Excellent

Quality of the paper: Is the overall quality of the paper suitable?

Good

Is the length of the paper justified?

Yes

Should the paper be seen by a specialist statistical reviewer?

No

Do you have any concerns about statistical analyses in this paper? If so, please specify them explicitly in your report.

Yes

It is a condition of publication that authors make their supporting data, code and materials available - either as supplementary material or hosted in an external repository. Please rate, if applicable, the supporting data on the following criteria.

Is it accessible?

Yes

Is it clear?

Yes

Is it adequate?

Yes

Do you have any ethical concerns with this paper?

No

Comments to the Author

Norman et al. conducted a very timely investigation of whether the primary visual cortex of blind individuals adheres to topographic mapping principles similar to retinotopic mapping in the sighted. They presented expert echolocators, blind non-echolocators and sighted controls with source and echo sounds from locations along the horizontal axis. They then tested if the fMRI activation for these sounds maps the space, as compared to predicted retinotopic mapping based on a probabilistic atlas, and whether it is correlated to the echolocation abilities.

The authors report that in echolocators (but not non-echolocating blind or sighted participants) primary visual cortex activation pattern mapping of the space correlated to the predicted retinotopic mapping. This is found for the echo stimuli, and to a lesser extent, for the source stimuli. Furthermore, they claim that the topographic mapping in the echolocators V1 is correlated to their behavioral abilities. The authors suggest this is evidence for task-specific organization in V1 in blindness, expanding this model of cortex organization to the early sensory cortices.

Overall the experimental design and writing are good, and the study addresses an important question in understanding brain plasticity and its organization. There are few analyses that need improved control to convincingly show the topographical mapping.

Major points:

1. Due to the protocol used (sparse sampling and continuous mapping, likely used to allow the subjects to hear echoes in the scanner), the spatiotopic maps are hard to judge visually. This is true for both the auditory echo/source maps in the blind and even for the retinotopic maps in the sighted. Therefore, readers must rely on the statistical analysis of the similarity of the maps to retinotopic from curvature correlations. Since this is the case, several improvements could make this analysis more robust:

a. The similarity analysis did not, as far as I could see, include a permutation test to show the chance level of correlation between the retinotopic probabilistic mapping based on anatomy and

the auditory spatiotopic map (from randomly assigning position for one of the maps). This would allow for statistical comparisons (based on the confidence interval) for the echolocators' topographic mapping, which is currently not present.

b. I found performing multiple correlations within groups of 5 subjects (to show the correlation of the retinotopic-like mapping with echo location ability) without correcting for multiple comparisons a bit troubling. It is certainly clear from the figures that there are 3 expert echolocators who show higher retinotopic-like mapping measures than the rest of the participants, but I'm not sure correlation is a convincing way to stress this point. It may be more prudent to show that these three individuals have both higher mapping scores and higher echolocating abilities as compared to the rest.

2. The authors showed, in the supplementary material, that the effect of contralaterality in the spatial representation is significant. However, it is unclear if the main measure of retinotopic-like mapping is significant because of the contralaterality effect alone (as may be the case in EE1) or because V1 also encodes for horizontal location within the hemifield. It could easily be tested by using the retinotopic probabilistic maps as two different models: one that only predicts the laterality and one that predicts the distance from the center (in absolute values). If both models are significant, it would go a long way towards making claims of detailed retinotopic-like mapping in the blind.

Minor points:

1. Another measure that can help demonstrate the reliability of these topographic maps is a test-retest reliability; specifically, the correlation of the echo maps to the source maps within individuals. Since these were taken on different days and with different inputs, this would be compelling evidence that these maps are reliable, and that they are found only in the echolocator experts.

2. The blind participants are quite variable in terms of their blindness onset, where some would be classified as late-onset blind (BC2, BC5), some have no clear onset of full blindness (EE3, BC2, BC4), and the experience with echolocation is variable in its onset as well. Certainly with echolocators one would not expect to exclude participants, but it would be beneficial to describe how these variable seem to affect to the findings.

3. I understand the theoretical focus of the authors on exploring V1, but it would also be very interesting to look beyond it in V2-V3. Given the direct connectivity between auditory cortex and early visual cortex (Beer A, Plank T, & Greenlee M (2011) Diffusion tensor imaging shows white matter tracts between human auditory and visual cortex. *Experimental Brain Research* 213(2):299-308; Cate AD, et al. (2009) Auditory attention activates peripheral visual cortex. *PLoS ONE* 4(2):e4645; Ungerleider LG & Desimone R (1986) Projections to the superior temporal sulcus from the central and peripheral field representations of V1 and V2. *The Journal of Comparative Neurology* 248(2):147-163), it could theoretically be that only V1 would show spatiotopic mapping, whereas V2-V3 would not. This could be interesting indication for the mechanism of cross-modal plasticity leading to task-specific organization in early sensory cortices. Moreover, it would be interesting to see if the LGN has a contralateral bias as well.

4. The authors state in their discussion that "we reason that primary 'visual' cortex is likely to be functionally necessary for the perception of space through sound, and in particular sound echoes, in some expert echolocators." (line 3-5, page 12). Given that they show fMRI data and not causal data (e.g. TMS) for the echolocators, it may be more cautious not to assume V1 is necessary.

5. The authors state that "One explanation for cross-modal reorganization of 'visual' cortex is that primary 'visual' cortex is more strongly connected to other sensory areas, including auditory

cortex" (lines 13-14 page 12). It may be worth noting that multiple studies have found decreased functional connectivity between the visual and auditory cortex in the blind as compared to controls (but not while performing auditory tasks: Pelland M, et al. (2017) State-dependent modulation of functional connectivity in early blind individuals. *NeuroImage* 147:532-541.), so a more careful phrasing may be advisable.

6. The authors may wish to mention, along with the studies they listed showing topographic connectivity patterns in the blind, also an older study which used TMS to address topography: Kupers R, et al. (2006) Transcranial magnetic stimulation of the visual cortex induces somatotopically organized qualia in blind subjects. *Proc Natl Acad Sci U S A* 103(35):13256-13260.

Review form: Reviewer 2

Recommendation

Major revision is needed (please make suggestions in comments)

Scientific importance: Is the manuscript an original and important contribution to its field?

Excellent

General interest: Is the paper of sufficient general interest?

Good

Quality of the paper: Is the overall quality of the paper suitable?

Good

Is the length of the paper justified?

No

Should the paper be seen by a specialist statistical reviewer?

No

Do you have any concerns about statistical analyses in this paper? If so, please specify them explicitly in your report.

No

It is a condition of publication that authors make their supporting data, code and materials available - either as supplementary material or hosted in an external repository. Please rate, if applicable, the supporting data on the following criteria.

Is it accessible?

No

Is it clear?

No

Is it adequate?

N/A

Do you have any ethical concerns with this paper?

No

Comments to the Author

My comments are in the attached file. (See Appendix A)

Decision letter (RSPB-2019-1333.R0)

25-Jul-2019

Dear Dr Norman:

I am writing to inform you that your manuscript RSPB-2019-1333 entitled "Retinotopic-like maps of spatial sound in primary 'visual' cortex of blind human echolocators" has, in its current form, been rejected for publication in Proceedings B.

This action has been taken on the advice of referees, who have recommended that substantial revisions are necessary. With this in mind we would be happy to consider a resubmission, provided the comments of the referees are fully addressed. However please note that this is not a provisional acceptance.

Sincerely,

Professor Gary Carvalho
mailto: proceedingsb@royalsociety.org

Associate Editor
Comments to Author:

Two expert reviewers have now seen your manuscript, and while both are very positive to the questions you are pursuing and the results you have obtained, they have significant reservations

about the experimental approach you have used in the study, the statistical analysis you have performed and the level of open access to the software and stimuli you have used. Both reviewers nonetheless think that you should be able to address their concerns and that a revised manuscript has the potential to have a significant impact on the field.

Reviewer(s)' Comments to Author:

Referee: 1

Comments to the Author(s)

Norman et al. conducted a very timely investigation of whether the primary visual cortex of blind individuals adheres to topographic mapping principles similar to retinotopic mapping in the sighted. They presented expert echolocators, blind non-echolocators and sighted controls with source and echo sounds from locations along the horizontal axis. They then tested if the fMRI activation for these sounds maps the space, as compared to predicted retinotopic mapping based on a probabilistic atlas, and whether it is correlated to the echolocation abilities.

The authors report that in echolocators (but not non-echolocating blind or sighted participants) primary visual cortex activation pattern mapping of the space correlated to the predicted retinotopic mapping. This is found for the echo stimuli, and to a lesser extent, for the source stimuli. Furthermore, they claim that the topographic mapping in the echolocators V1 is correlated to their behavioral abilities. The authors suggest this is evidence for task-specific organization in V1 in blindness, expanding this model of cortex organization to the early sensory cortices.

Overall the experimental design and writing are good, and the study addresses an important question in understanding brain plasticity and its organization. There are few analyses that need improved control to convincingly show the topographical mapping.

Major points:

1. Due to the protocol used (sparse sampling and continuous mapping, likely used to allow the subjects to hear echoes in the scanner), the spatiotopic maps are hard to judge visually. This is true for both the auditory echo/source maps in the blind and even for the retinotopic maps in the sighted. Therefore, readers must rely on the statistical analysis of the similarity of the maps to retinotopic from curvature correlations. Since this is the case, several improvements could make this analysis more robust:

a. The similarity analysis did not, as far as I could see, include a permutation test to show the chance level of correlation between the retinotopic probabilistic mapping based on anatomy and the auditory spatiotopic map (from randomly assigning position for one of the maps). This would allow for statistical comparisons (based on the confidence interval) for the echolocators' topographic mapping, which is currently not present.

b. I found performing multiple correlations within groups of 5 subjects (to show the correlation of the retinotopic-like mapping with echo location ability) without correcting for multiple comparisons a bit troubling. It is certainly clear from the figures that there are 3 expert echolocators who show higher retinotopic-like mapping measures than the rest of the participants, but I'm not sure correlation is a convincing way to stress this point. It may be more prudent to show that these three individuals have both higher mapping scores and higher echolocating abilities as compared to the rest.

2. The authors showed, in the supplementary material, that the effect of contralaterality in the spatial representation is significant. However, it is unclear if the main measure of retinotopic-like mapping is significant because of the contralaterality effect alone (as may be the case in EE1) or because V1 also encodes for horizontal location within the hemifield. It could easily be tested by using the retinotopic probabilistic maps as two different models: one that only predicts the laterality and one that predicts the distance from the center (in absolute values). If both models are significant, it would go a long way towards making claims of detailed retinotopic-like mapping in the blind.

Minor points:

1. Another measure that can help demonstrate the reliability of these topographic maps is a test-retest reliability; specifically, the correlation of the echo maps to the source maps within individuals. Since these were taken on different days and with different inputs, this would be compelling evidence that these maps are reliable, and that they are found only in the echolocator experts.

2. The blind participants are quite variable in terms of their blindness onset, where some would be classified as late-onset blind (BC2, BC5), some have no clear onset of full blindness (EE3, BC2, BC4), and the experience with echolocation is variable in its onset as well. Certainly with echolocators one would not expect to exclude participants, but it would be beneficial to describe how these variable seem to affect to the findings.

3. I understand the theoretical focus of the authors on exploring V1, but it would also be very interesting to look beyond it in V2-V3. Given the direct connectivity between auditory cortex and early visual cortex (Beer A, Plank T, & Greenlee M (2011) Diffusion tensor imaging shows white matter tracts between human auditory and visual cortex. *Experimental Brain Research* 213(2):299-308; Cate AD, et al. (2009) Auditory attention activates peripheral visual cortex. *PLoS ONE* 4(2):e4645; Ungerleider LG & Desimone R (1986) Projections to the superior temporal sulcus from the central and peripheral field representations of V1 and V2. *The Journal of Comparative Neurology* 248(2):147-163), it could theoretically be that only V1 would show spatiotopic mapping, whereas V2-V3 would not. This could be interesting indication for the mechanism of cross-modal plasticity leading to task-specific organization in early sensory cortices. Moreover, it would be interesting to see if the LGN has a contralateral bias as well.

4. The authors state in their discussion that “we reason that primary ‘visual’ cortex is likely to be functionally necessary for the perception of space through sound, and in particular sound echoes, in some expert echolocators.” (line 3-5, page 12). Given that they show fMRI data and not causal data (e.g. TMS) for the echolocators, it may be more cautious not to assume V1 is necessary.

5. The authors state that “One explanation for cross-modal reorganization of ‘visual’ cortex is that primary ‘visual’ cortex is more strongly connected to other sensory areas, including auditory cortex” (lines 13-14 page 12). It may be worth noting that multiple studies have found decreased functional connectivity between the visual and auditory cortex in the blind as compared to controls (but not while performing auditory tasks: Pelland M, et al. (2017) State-dependent modulation of functional connectivity in early blind individuals. *NeuroImage* 147:532-541.), so a more careful phrasing may be advisable.

6. The authors may wish to mention, along with the studies they listed showing topographic connectivity patterns in the blind, also an older study which used TMS to address topography:

Kupers R, et al. (2006) Transcranial magnetic stimulation of the visual cortex induces somatotopically organized qualia in blind subjects. Proc Natl Acad Sci U S A 103(35):13256-13260.

Referee: 2

Comments to the Author(s)

My comments are in the attached file.

Author's Response to Decision Letter for (RSPB-2019-1333.R0)

See Appendix B.

RSPB-2019-1910.R0

Review form: Reviewer 2

Recommendation

Accept as is

Scientific importance: Is the manuscript an original and important contribution to its field?

Excellent

General interest: Is the paper of sufficient general interest?

Good

Quality of the paper: Is the overall quality of the paper suitable?

Good

Is the length of the paper justified?

Yes

Should the paper be seen by a specialist statistical reviewer?

No

Do you have any concerns about statistical analyses in this paper? If so, please specify them explicitly in your report.

No

It is a condition of publication that authors make their supporting data, code and materials available - either as supplementary material or hosted in an external repository. Please rate, if applicable, the supporting data on the following criteria.

Is it accessible?

Yes

Is it clear?

Yes

Is it adequate?

Yes

Do you have any ethical concerns with this paper?

No

Comments to the Author

The authors have worked on the revisions thoroughly and clarified the raised points. I am convinced that the work has a significant importance in the field and I strongly endorse it for the publication.

Decision letter (RSPB-2019-1910.R0)

10-Sep-2019

Dear Dr Norman

I am pleased to inform you that your Review manuscript RSPB-2019-1910 entitled "Retinotopic-like maps of spatial sound in primary 'visual' cortex of blind human echolocators" has been accepted for publication in Proceedings B.

The referee(s) do not recommend any further changes. Therefore, please proof-read your manuscript carefully and upload your final files for publication. Because the schedule for publication is very tight, it is a condition of publication that you submit the revised version of your manuscript within 7 days. If you do not think you will be able to meet this date please let me know immediately.

To upload your manuscript, log into <http://mc.manuscriptcentral.com/prsb> and enter your Author Centre, where you will find your manuscript title listed under "Manuscripts with Decisions." Under "Actions," click on "Create a Revision." Your manuscript number has been appended to denote a revision.

You will be unable to make your revisions on the originally submitted version of the manuscript. Instead, upload a new version through your Author Centre.

- 1) A text file of the manuscript (doc, txt, rtf or tex), including the references, tables (including captions) and figure captions. Please remove any tracked changes from the text before submission. PDF files are not an accepted format for the "Main Document".
- 2) A separate electronic file of each figure (tiff, EPS or print-quality PDF preferred). The format should be produced directly from original creation package, or original software format. Please note that PowerPoint files are not accepted.

3) Electronic supplementary material: this should be contained in a separate file from the main text and the file name should contain the author's name and journal name, e.g. `authorname_procb_ESM_figures.pdf`

All supplementary materials accompanying an accepted article will be treated as in their final form. They will be published alongside the paper on the journal website and posted on the online figshare repository. Files on figshare will be made available approximately one week before the accompanying article so that the supplementary material can be attributed a unique DOI. Please see: <https://royalsociety.org/journals/authors/author-guidelines/>

4) Data-Sharing and data citation

It is a condition of publication that data supporting your paper are made available. Data should be made available either in the electronic supplementary material or through an appropriate repository. Details of how to access data should be included in your paper. Please see <https://royalsociety.org/journals/ethics-policies/data-sharing-mining/> for more details.

<http://datadryad.org/submit?journalID=RSPB&manu=RSPB-2019-1910> which will take you to your unique entry in the Dryad repository.

Once again, thank you for submitting your manuscript to Proceedings B and I look forward to receiving your final version. If you have any questions at all, please do not hesitate to get in touch.

Sincerely,

Professor Gary Carvalho
<mailto:proceedingsb@royalsociety.org>

Associate Editor,
 Board Member
 Comments to Author:

One of the previous reviewers has now seen your latest submission and is entirely satisfied with the efforts you have made to address the comments and criticisms that were raised following your earlier submission. This reviewer feels that your manuscript requires no further revision and has recommended that it be accepted as is.

Reviewer(s)' Comments to Author:

Referee: 2

Comments to the Author(s).

The authors have worked on the revisions thoroughly and clarified the raised points. I am convinced that the work has a significant importance in the field and I strongly endorse it for the publication.

Decision letter (RSPB-2019-1910.R1)

12-Sep-2019

Dear Dr Norman

I am pleased to inform you that your manuscript entitled "Retinotopic-like maps of spatial sound in primary 'visual' cortex of blind human echolocators" has been accepted for publication in Proceedings B.

Open Access

You are invited to opt for Open Access, making your freely available to all as soon as it is ready for publication under a CCBY licence. Our article processing charge for Open Access is £1700. Corresponding authors from member institutions (<http://royalsocietypublishing.org/site/librarians/allmembers.xhtml>) receive a 25% discount to these charges. For more information please visit <http://royalsocietypublishing.org/open-access>.

Paper charges

Sincerely,

Proceedings B
mailto: proceedingsb@royalsociety.org

Appendix A

GENERAL MARKS

In the current study, Norman & Thaler aimed at assessing retinotopy-like maps in the visual primary cortex of blind echo-locators, the blind individuals without echo localization abilities and sighted controls. They acquired sparse sampling fMRI data to map neural responses to 8 horizontal spatial eccentricities both echo sounds and source sounds (and also in vision in sighted participants), then they related the degree of retinotopy-like mapping values with each individuals' echo location abilities. They observed that: (a) degree of retinotopy-like mapping of echo sounds significantly link to the echo location expertise; (b) better echo location ability in the blind echo locator experts primary visual cortex is more likely to show retinotopy-like mapping; (c) the same pattern of results were observed for degree of contralateral mapping of echo sounds; (d) the degree of sound source retinotopic-like mapping is associated with echolocation expertise, however, echo localization ability did not relate to the sound source maps in none of the participant groups.

I read this paper with great interest. The topic of investigation is timely and still open to discussion: can we find evidence of task-specific organization in the primary sensory regions after functional reorganization in sensory deprived individuals?

While the manuscript appears well organized and the analyses are good, other theoretical and methodological aspects make the reported main observations debatable.

MAJOR

INTRODUCTION

Early visual cortex (EVC) is known to be activated by non-visual task in congenitally or early blind individuals. Could the authors explain better why they would not predict, at least, sound source specific activity in the EVC? If not, what is the main difference in processing spatial sounds between expert blind echolocators and early blind individuals?

The authors have not mentioned what are the definitions of retinotopy-like maps. Could they explain further what the criteria are to confirm if there is a topographic organization or not.

MATERIAL & METHODS

The choice of the fMRI acquisition protocol is difficult to understand considering the questions authors aiming to address. The retinotopy maps require continuous stimulation. In the sighted controls visual maps,

do not seem to be retinotopically organized, there is not strong contralateral preference as also in the correlation to an atlas shows $r=0.59$. Is sparse-sampling technique advantageous for retinotopy-like mapping? Could the low variance-explained/coherence threshold in the data due to the acquisition protocol?

Can the authors be more explicit about why did they choose the correlation coefficient as the measure of behavioral performance? How did they perform the correlation, vector-wise? What does correlation vector-wise mean? Does correlation between the prediction and observe value a good indicator of spatial localization ability? If they had chosen the accuracy as a measure, would they observe the same pattern of results? I suspect the accuracy measure may not be fine-tune, then their experimental design is also a good fit for psychometric curve fitting for instance, and the threshold (extracted from each individuals' curve) could be a good measure for ability of localization.

How reliable the correlation analysis, considering the subject pool being very small, $n=5$?

It is not clear to me that why authors did not correlate the obtained sighted control visual retinotopy-like maps with auditory maps. Instead, they relied on Benson atlas as a measure. What are the reasons to use external atlas and then discuss/conclude the way they do in the Discussion (page 10, line 23-24). I suspect they wanted to maintain the same analysis across 3 groups. However, in order to meet their claims in the Discussion, I would suggest showing the same pattern of results when the degree of retinotopy-like maps emerging from correlation between visual and auditory maps.

Related to the low coherence threshold, the authors have mentioned the upper limit, due to the visual retinotopy-like mapping. In order to distinguish the observed retinotopy-like mapping in the echolocators for echo-sounds, it would be crucial to have lower limit for each individual's maps. There are no whole-brain results, so it is hard to understand how much of the results are noise or task-related. One way to test this, is to show whole brain results, and probably taking region-of-no-interest coherence. The coherence in that region could be a threshold/baseline. Then assigning this value as a "baseline" and/or using to perform stats to understand each individual's' variance explain/coherence is significantly different than the one observed in no-interest region (noise). Another way would be splitting the data into two halves and running correlation to show that the observation is different from noise.

The cross-correlation results are within the mask of V1, V2, V3? Is there a measure for that, e.g. FreeSurfer based? I'm guessing it's from Benson atlas, they masked their functional results with. Nevertheless, it's not clear in the manuscript. A way would be representing in the Figure 1 the landmark of Benson atlas.

It would help the reader to understand the perceptual difference between auditory echo localization and sound source localization if the stimuli were also provided. For the sake of replicability of the analysis, and clarity of the technique that was used, it is essential to have open access to the custom Matlab/R scripts. E.g. which function do they use for multiple linear regression analysis, are assumptions for linear regression analysis met? The correlation values are normally distributed? Please see the simple R script in the link as an example.

https://www.dropbox.com/s/7hsctigt2rbgd82/LM_assumption_checks.R?dl=0

Can they explain briefly why they chose for multiple regression analysis and specifically that model?

When multiple regression analyses results are reported the following items were missing: effect sizes, confidence intervals, etc... Inserting variance explained value right after the model parameters/regressors definitions, would potentially avoid the confusion. Is the variance explained here R^2 ?

RESULTS

How do authors explain the sound sources evoked less response/location preference compared to the echo sounds (i.e. SC2, SC3, SC4)? I'm guessing that stimuli energy levels across two auditory tasks are not equalized to be ecologically valid. Do the auditory stimuli of the 8 spatial locations show the same frequency? Would frequency or energy difference across stimuli explain their results? The click emission is not broadband stimulus, might explain the weak activity in sighted and blind participants? What are the differences between stimuli (echo vs. source sounds)?

Were authors expecting ipsilateral response in the visual retinotopy experiment in sighted controls? Can they elaborate what could it mean? Did the results show contralaterality in the visual cortex? Maybe the authors had performed such analysis, but it is not clear in the manuscript. Such analyses would increase the reliability of the results, since with visual inspection some sighted controls seems to not follow "typical" retinotopic organization.

The degree of sound source retinotopy-like map was correlated with echo localization ability instead investigating correlations with sound localization ability. What was the hypothesis for this analysis?

Figures: in such inflated brain, the curvatures are not visible, please consider smoothing less. Also it would be better for the reader to see Figure 1 and 2 together to compare the maps visually. Furthermore, without

visual inspection of the sighted controls or blind individuals' maps, I find the results are not informative in the main text. I would highly recommend adding at least one of the two groups' results into main manuscript.

By visual inspection, EE3 and EE4 are the two participants showing not consistent maps across echo sounds and source sounds, especially in the right hemisphere, while three other subjects showing consistent/similar maps for both echo and source sounds. Is there any hypothesis for such results?

DISCUSSION

Considering echolocator experts show retinotopy-like mapping for echo sounds, what is the explanation or theoretical framework of weaker sound-source evoked retinotopy-like maps in the EVC of echolocators? Could the authors discuss briefly what are possible explanations of the observed differences in echo localization vs. sound source localization in the brain?

Page 10, line 21-24, the authors claimed that retinotopic map in the EVC can be used to map sensory non-visual input. However, there is no analysis to show that retinotopy maps in the visually deprived individuals stem from the typically retinotopic organization, e.g. assuming the maps are located only in V1, are the zones in the visual maps showing contralateral response showing same/similar preference in the auditory maps? The inter-subject variability seems to be high, even in auditory tasks only, same zones seem to not show same preference of space in the blind individuals (see Material method section, point 4, suggestion for performing correlation analysis only in the sighted group to investigate whether same voxels showing XX eccentricity or contra-laterality preference in the visual maps, would show similar preference in auditory locations?).

The results also showed little evidence to retinotopy-like mapping in blind individuals both for echo and source sounds. Considering the importance of spatial perception in blind individuals, it seems rather surprising that early blind participants did not show retinotopy-like maps in the visual cortex. What could be the explanation? Could it be due to low number of participants? And/or the fMRI acquisition protocol? Considering the previous studies in the field focused on the early blind individuals, and how non-visual input functionally mapped on the visual cortex, how do the authors explain their results (Page 11, line 9-20).

Page 12, line 4-5, 16-17, could the authors further discuss the differences in the processing of spatial localization in echo-locators and non-echolocator blind individuals?

MINOR

Introduction

Page 3, line 8-10, 17-19 need more clarity, clear transition and/or further explanation.

Results

Page 5, line 16-23 needs more clarification.

Discussion

Page 12, line 4-5, can authors cite more studies to support their prediction? According to their prediction/claim, one could also predict that early blinds could show map-like organization for spatial perception in EVC.

Appendix B

Please find responses (in red) to each of the reviewer's comments below:

Associate Editor

Comments to Author:

Two expert reviewers have now seen your manuscript, and while both are very positive to the questions you are pursuing and the results you have obtained, they have significant reservations about the experimental approach you have used in the study, the statistical analysis you have performed and the level of open access to the software and stimuli you have used. Both reviewers nonetheless think that you should be able to address their concerns and that a revised manuscript has the potential to have a significant impact on the field.

We thank the reviewers for their constructive feedback. In our revision we have addressed their comments and taken up many of their suggestions.

Below we detail how we addressed their comments. Page and line numbers refer to the revised manuscript with changes tracked.

Reviewer(s)' Comments to Author:

Referee: 1

Comments to the Author(s)

Norman et al. conducted a very timely investigation of whether the primary visual cortex of blind individuals adheres to topographic mapping principles similar to retinotopic mapping in the sighted. They presented expert echolocators, blind non-echolocators and sighted controls with source and echo sounds from locations along the horizontal axis. They then tested if the fMRI activation for these sounds maps the space, as compared to predicted retinotopic mapping based on a probabilistic atlas, and whether it is correlated to the echolocation abilities.

The authors report that in echolocators (but not non-echolocating blind or sighted participants) primary visual cortex activation pattern mapping of the space correlated to the predicted retinotopic mapping. This is found for the echo stimuli, and to a lesser extent, for the source stimuli. Furthermore, they claim that the topographic mapping in the echolocators V1 is correlated to their behavioral abilities. The authors suggest this is evidence for task-specific organization in V1 in blindness, expanding this model of cortex organization to the early sensory cortices.

Overall the experimental design and writing are good, and the study addresses an important question in understanding brain plasticity and its organization. There are few analyses that need improved control to convincingly show the topographical mapping.

Major points:

1. Due to the protocol used (sparse sampling and continuous mapping, likely used to allow the subjects to hear echoes in the scanner), the spatiotopic maps are hard to judge visually. This is true for both the auditory echo/source maps in the blind and even for the retinotopic maps in the sighted. Therefore, readers must rely on the statistical analysis of the similarity of the maps to retinotopic from curvature correlations. Since this is the case, several improvements could make this analysis more robust:

a. The similarity analysis did not, as far as I could see, include a permutation test to show the chance level of correlation between the retinotopic probabilistic mapping based on anatomy and the auditory spatiotopic map (from randomly assigning position for one of the maps). This would allow for statistical comparisons (based on the confidence interval) for the echolocators' topographic mapping, which is currently not present.

We would like to thank the reviewer for their suggestion.

To address their concerns we have now included 95% CI around each correlation between measured maps and probabilistic maps expected based on anatomy (Benson atlas) reported in the manuscript. These data are now included in Supplementary table S1 and are referred to in the main text:

"These correlation coefficients, along with 95% confidence limits (derived from a bootstrapping method), are shown in Supplementary Table S1." Page 6, line 8

95% confidence limits were computed by randomly resampling the data used to compute a correlation 1000 times (using matlab's *bootstrp* function) and each time calculating a correlation coefficient for the resampled dataset. Subsequently we calculated standard error of the mean on these bootstrapped statistics, and then reported 95% confidence limits in table S1.

These data show that the degree of retinotopic-like mapping of echo sounds in the 3 highest-performing expert echolocators is above those acquired for the control participants.

Furthermore, as suggested by the reviewer, to obtain an idea of what chance level of correlation would be between measured maps and probabilistic maps expected based on anatomy (Benson atlas), we also implemented the suggestion by reviewer 1, i.e. we randomly 'scrambled' measured maps 1000 times and each time calculated the correlation between the scrambled map and the atlas based map. This allowed us to obtain an idea what chance performance would be, as well as 95% CI around these. These values are shown in supplementary table S2 and referred to in the main text:

"We also confirmed that correlation coefficients expected by chance are indeed zero (results also shown in Supplementary Table S2)." Page 6, line 9

These data indicate an extremely low level of chance performance (around 0.00), with very small confidence intervals (around +/- 0.02) across all participants. This suggests that our measures of retinotopic-like mapping are not the result of chance.

b. I found performing multiple correlations within groups of 5 subjects (to show the correlation of the retinotopic-like mapping with echo location ability) without correcting for multiple comparisons a bit troubling. It is certainly clear from the figures that there are 3 expert echolocators who show higher retinotopic-like mapping measures than the rest of the participants, but I'm not sure correlation is a convincing way to stress this point. It may be more prudent to show that these three individuals have both higher mapping scores and higher echolocating abilities as compared to the rest.

We would like to thank the reviewer for this suggestion.

To address this we have now additionally included statistical analyses that test whether any of the 5 expert echolocators have significantly greater retinotopic mapping, compared to the 10 control participants (blind and sighted controls collapsed). These tests are done using modified t-tests

(Crawford & Howell, 1998; Crawford & Garthwaite, 2002), which test whether a single case differs significantly from a control group.

The results show that, for mapping of echo sounds, three of the 5 echolocators have significantly more retinotopic-like mapping than controls and, for the mapping of source sounds, one of the 5 echolocators does.

These statistics are included in the supplementary materials and referred to in the main text:

Echo sounds:

“Separate single-case statistics [26, 27] confirm that EEs 1, 3 and 5 have a significantly higher degree of retinotopic-like mapping of echoes compared to controls (see Supplementary Materials).” Page 12, line 14

Source sounds:

“Separate single-case statistics [26, 27] confirm that EE5 has a significantly higher degree of retinotopic-like mapping of source sounds compared to controls (see Supplementary Materials).” Page 14, line 20

We have also carried out the same analysis for the behavioural performance data. We find that all 5 echolocators have higher echo localisation ability compared to controls. For source sound localisation, none of the echolocators is better than the controls (but this is expected, as all participants performed very well on this task). This is reported in the supplementary material, and referred to in the main text:

“As expected, EEs had better echo localisation ability compared to controls, and all participants were very good at localising source sounds (see Supplementary Materials text and Supplementary Table S5).” Page 11, line 4

To a degree the analysis in response to reviewer 1’s first major concern (i.e. 95% CI intervals) also addresses this.

The multiple linear regression analysis for the whole group (and follow up regressions for subgroups) allows us to include all predictors in one analysis (i.e. blindness, expertise, performance, interaction), which is why we kept this analysis in the manuscript. The MLR controls for type I errors within the analysis by adjusting the degrees of freedom.

2. The authors showed, in the supplementary material, that the effect of contralaterality in the spatial representation is significant. However, it is unclear if the main measure of retinotopic-like mapping is significant because of the contralaterality effect alone (as may be the case in EE1) or because V1 also encodes for horizontal location within the hemifield. It could easily be tested by using the retinotopic probabilistic maps as two different models: one that only predicts the laterality and one that predicts the distance from the center (in absolute values). If both models are significant, it would go a long way towards making claims of detailed retinotopic-like mapping in the blind.

Based on the reviewer’s suggestion we now provide additional analyses that investigate eccentricity separately from laterality. Specifically, we ran two separate linear regression models on the mapping data for each of the three EEs who had shown retinotopic like mapping for echoes (compared to controls) – one that used the retinotopic atlas as a predictor only of laterality (i.e. all values in the atlas and in the observed maps were converted to either -1 or 1 to represent left and right space, respectively), and one that used the atlas as a predictor only of eccentricity (i.e. the absolute values of all values in the atlas and in the observed maps were taken to represent the

distance from central space). We find that the effect of laterality is significant in all three of these EEs, and eccentricity is significant in two of them (EE3 and EE5). This suggests that, at least in these two EEs, primary visual cortex maps the horizontal position of echo sounds as well as laterality. We also include the same analysis for source sound localisation in one of the EEs (EE5, who was the only one to have statistically more retinotopic-like mapping compared to controls) and find evidence of mapping of both contralaterality and eccentricity.

We have included this analysis in the supplemental material and refer to it in the main text:

Echo sounds:

“Further analyses showed that this mapping in EE3 and EE5 can explained by mapping of both laterality and eccentricity, whilst for EE1 the effect is driven by laterality (see Supplementary Materials).” Page 12, line 8

Source sounds:

“Further analyses showed that this mapping in EE5 can explained by mapping of both laterality and eccentricity (see Supplementary Materials).” Page 14, line 6.

Minor points:

1. Another measure that can help demonstrate the reliability of these topographic maps is a test-retest reliability; specifically, the correlation of the echo maps to the source maps within individuals. Since these were taken on different days and with different inputs, this would be compelling evidence that these maps are reliable, and that they are found only in the echolocator experts.

In order to demonstrate the reliability of maps, we have included 95% CI around measured correlations based on boot strapping (see also our response to comment 1a).

In addition, we also include an analysis that compares the maps in V1 with those in V2 and V3. We find that there is a strong positive correlation between the degree of retinotopic-like mapping observed in V1 and in V2 and V3, for both echo and source sounds. This strongly suggests that the observed neural maps are reliable and not due to low statistical power.

We decided to use these approaches rather than comparing echo and source maps, due to there being no *a priori* theory to expect that echo and source maps will in fact be the same.

We have included these additional analysis and results in the supplemental material and refer to them in the main text:

“Whilst for theoretical reasons the main focus of our manuscript is on V1, we also carried out the same mapping analyses in the second and third visual cortex (V2 and V3), each of which contain a retinotopically organised map of visual space [28] and are directly connected to auditory cortex [29-31]. We also measured similarity of maps in V2 and V3 to those observed in V1. For those EEs where we found retinotopic-like mapping of echo and source sounds in primary ‘visual’ cortex, we also find it in V2 and V3, suggesting that retinotopic-like mapping of echo and source sounds extends beyond the primary ‘visual’ area. Furthermore, high similarity between maps in V1 and V2/V3 strongly suggest that maps in primary ‘visual’ cortex are reliable and not the result of low statistical power. All of this is reported in the Supplementary Materials.” Page 14, line 8

Mapping data for V2 and V3 are shown in supplementary tables S3 and S4, respectively.

2. The blind participants are quite variable in terms of their blindness onset, where some would be classified as late-onset blind (BC2, BC5), some have no clear onset of full blindness (EE3, BC2, BC4), and the experience with echolocation is variable in its onset as well. Certainly with echolocators one would not expect to exclude participants, but it would be beneficial to describe how these variables seem to affect to the findings.

We agree that BC5 is classified as late blind, whilst EE3, BC2 and BC4 (whilst all early blind) have no clear onset of full blindness. We also agree that the experience in echolocation is variable in its onset. The effect that these variables may play is an important point to discuss, and we thank the reviewer for raising this issue. We have now expanded part of the relevant discussion section to address this:

“It is clear from the results of the present study, however, that blindness is not sufficient to explain the cross-modal recruitment of primary ‘visual’ cortex, as blind control participants did not show evidence of retinotopic-like mapping either for source or echo sounds. It is important to note, however, that there was variation in the onset of blindness in our sample, i.e. BC5 is classified as late blind and EE3, BC2 and BC4 (whilst early blind) have no clear onset of total blindness, and that also the age at which EEs started using echolocation is variable. However, there was no relationship between degree of retinotopic-like mapping and age at onset of blindness in either group, and in our EE sample there was also no relationship between neural mapping and age at onset of echolocation use. Furthermore, because the degree of retinotopic-like mapping of sound in the expert echolocators was positively associated with echo localisation ability, proficiency in processing sensory input might be the critical factor in the neural reorganisation of primary sensory areas. This is in line with previous studies showing cross-modal activity in early ‘visual’ cortex from proficient sensory skill use (e.g. Braille, or sensory substitution devices [10]; monaural sound localization [12]).” Page 16, line 23

3. I understand the theoretical focus of the authors on exploring V1, but it would also be very interesting to look beyond it in V2-V3. Given the direct connectivity between auditory cortex and early visual cortex (Beer A, Plank T, & Greenlee M (2011) Diffusion tensor imaging shows white matter tracts between human auditory and visual cortex. *Experimental Brain Research* 213(2):299-308; Cate AD, et al. (2009) Auditory attention activates peripheral visual cortex. *PLoS ONE* 4(2):e4645; Ungerleider LG & Desimone R (1986) Projections to the superior temporal sulcus from the central and peripheral field representations of V1 and V2. *The Journal of Comparative Neurology* 248(2):147-163), it could theoretically be that only V1 would show spatiotopic mapping, whereas V2-V3 would not. This could be interesting indication for the mechanism of cross-modal plasticity leading to task-specific organization in early sensory cortices. Moreover, it would be interesting to see if the LGN has a contralateral bias as well.

We thank the reviewer for their suggestions.

For theoretical reasons the focus is on V1. But, to address the points of the reviewer, which other readers are also likely to think about, we have now run the same analysis that we did for V1, also for V2 and V3. We refer to these results in the main manuscript and report them in the supplemental material Tables S2 and S3.

In the main manuscript we now write:

“Whilst for theoretical reasons the main focus of our manuscript is on V1, we also carried out the same mapping analyses in the second and third visual cortex (V2 and V3), each of which contain a retinotopically organised map of visual space [28] and are directly connected to auditory cortex [29-31]. We also measured similarity of maps in V2 and V3 to those observed in V1. For those EEs where we found retinotopic-like mapping of echo and source sounds in primary ‘visual’ cortex, we also find it in V2 and V3, suggesting that retinotopic-like mapping of echo and source sounds extends beyond the primary ‘visual’ area. Furthermore, high similarity between maps in V1 and V2/V3 strongly suggest that maps in primary ‘visual’ cortex are reliable and not the result of low statistical power. All of this is reported in the Supplementary Materials.” Page 14, line 8

4. The authors state in their discussion that “we reason that primary ‘visual’ cortex is likely to be functionally necessary for the perception of space through sound, and in particular sound echoes, in some expert echolocators.” (line 3-5, page 12). Given that they show fMRI data and not causal data (e.g. TMS) for the echolocators, it may be more cautious not to assume V1 is necessary.

Yes.

We have now included a more cautious wording in the discussion:

“Given that previous studies have shown that functionally reorganised visual areas are in fact causally involved in processing non-visual information – e.g. through the application of neurostimulation [32 – 33] – it is possible that primary ‘visual’ cortex is functionally necessary for the perception of space through sound, and in particular sound echoes, in some expert echolocators.” Page 16, line 8

5. The authors state that “One explanation for cross-modal reorganization of ‘visual’ cortex is that primary ‘visual’ cortex is more strongly connected to other sensory areas, including auditory cortex” (lines 13-14 page 12). It may be worth noting that multiple studies have found decreased functional connectivity between the visual and auditory cortex in the blind as compared to controls (but not while performing auditory tasks: Pelland M, et al. (2017) State-dependent modulation of functional connectivity in early blind individuals. *NeuroImage* 147:532-541.), so a more careful phrasing may be advisable.

We would like to thank the reviewer for this helpful suggestion. We have now included a statement in the discussion that mentions this:

“These greater structural connections might confer additional functional connectivity between auditory and visual areas when an individual performs a relevant auditory task, but not necessarily at rest [36].” Page 16, line 21

6. The authors may wish to mention, along with the studies they listed showing topographic connectivity patterns in the blind, also an older study which used TMS to address topography: Kupers R, et al. (2006) Transcranial magnetic stimulation of the visual cortex induces somatotopically organized qualia in blind subjects. *Proc Natl Acad Sci U S A* 103(35):13256-13260.

We thank the reviewer for the suggestion of this study. We have included a reference to this study in the introduction.

We now write:

“There is also one previous study that used TMS to map the topographical representation of somatosensory input in visual cortex [18], but the cortical sites stimulated in that study covered many visual association areas in addition to primary visual cortex. The presence of a cross-modal

topographical stimulus map within primary visual cortex, therefore, remains to be shown.” Page 3, line 19

Reviewer 2

GENERAL MARKS

In the current study, Norman & Thaler aimed at assessing retinotopy-like maps in the visual primary cortex of blind echo-locators, the blind individuals without echo localization abilities and sighted controls. They acquired sparse sampling fMRI data to map neural responses to 8 horizontal spatial eccentricities both echo sounds and source sounds (and also in vision in sighted participants), then they related the degree of retinotopy-like mapping values with each individuals' echo location abilities. They observed that: (a) degree of retinotopy-like mapping of echo sounds significantly link to the echo location expertise; (b) better echo location ability in the blind echo locator experts primary visual cortex is more likely to show retinotopy-like mapping; (c) the same pattern of results were observed for degree of contralateral mapping of echo sounds; (d) the degree of sound source retinotopic-like mapping is associated with echolocation expertise, however, echo localization ability did not relate to the sound source maps in none of the participant groups. I read this paper with great interest. The topic of investigation is timely and still open to discussion: can we find evidence of task-specific organization in the primary sensory regions after functional reorganization in sensory deprived individuals? While the manuscript appears well organized and the analyses are good, other theoretical and methodological aspects make the reported main observations debatable.

MAJOR

INTRODUCTION

Early visual cortex (EVC) is known to be activated by non-visual task in congenitally or early blind individuals.

Could the authors explain better why they would not predict, at least, sound source specific activity in the EVC? If not, what is the main difference in processing spatial sounds between expert blind echolocators and early blind individuals?

We would like to thank the reviewer for pointing out that we do need to explain more carefully.

We would first like to clarify that our neural data are the result of a cross correlation analysis, and as such they do not represent the strength of activity in response to auditory stimulation per se, but rather how consistently a specific voxel responded to a specific stimulus position. Thus, these data should not be taken as indicators of the strength of activity in primary visual cortex to sound, as you could for example measure in an experimental design that would compare responses to acoustic stimulation to responses in a non-acoustic control condition.

We have added a section in the results to clarify this:

“It is important to note that neural data are the result of a cross correlation analysis, and as such they do not represent the strength of activity in response to auditory stimulation per se, but rather how consistently a specific voxel responded to a specific stimulus position.” Page 5, line 8.

Nonetheless, it is important for us to clarify our interpretation why we observed more consistent mapping for echo sounds as compared to source sounds.

As outlined in the introduction, we used echolocation as a perceptual paradigm alongside source localization because in this way the inclusion of blind echolocators as well as blind non-echolocators enabled us to investigate the effects of blindness and expertise simultaneously (in this case: expertise in echolocation). Any retinotopic-like mapping in V1 in any condition is in principle consistent with task specificity in V1 (rather than modality specificity).

Our results show that mapping of echo sounds in V1 is associated with echo expertise. They also show that mapping of source sounds in V1 is associated with echo expertise, though to a lesser degree (i.e. less explained variance). Whilst this clearly demonstrates task specificity (rather than modality specificity) in V1, it also suggests that this task specificity does not arise by default in response to blindness (as in that case every blind participant should have shown mapping for sound sources as everyone can perceive them well), but that for this task-specific adaptation of V1 proficiency in processing sensory input (here: echolocation expertise) might be the critical factor. As we state in our discussion, this finding and interpretation is in line with previous studies showing cross-modal activity in early 'visual' cortex from proficient sensory skill use (e.g. Braille, or sensory substitution devices; monaural sound localization).

Also in response to a comment from reviewer 1 (which we think is related) we now write in the discussion:

“It is clear from the results of the present study, however, that blindness is not sufficient to explain the cross-modal recruitment of primary ‘visual’ cortex, as blind control participants did not show evidence of retinotopic-like mapping either for source or echo sounds. It is important to note, however, that there was variation in the onset of blindness in our sample, i.e. BC5 is classified as late blind and EE3, BC2 and BC4 (whilst early blind) have no clear onset of total blindness, and that also the age at which EEs started using echolocation is variable. However, there was no relationship between degree of retinotopic-like mapping and age at onset of blindness in either group, and in our EE sample there was also no relationship between neural mapping and age at onset of echolocation use. Furthermore, because the degree of retinotopic-like mapping of sound in the expert echolocators was positively associated with echo localisation ability, proficiency in processing sensory input might be the critical factor in the neural reorganisation of primary sensory areas. This is in line with previous studies showing cross-modal activity in early ‘visual’ cortex from proficient sensory skill use (e.g. Braille, or sensory substitution devices [10]; monaural sound localization [12]).” Page 16, line 23

The finding that in those individuals who did show mapping, we observed weaker mapping for source sounds, as compared to echo sounds, might be explained by the fact that the typical brain areas that are responsible for spatially mapping source sounds are still intact in all our participants, and so there is perhaps less need for the brain to undergo neural reorganisation in order to map these in V1.

To make this very clear in our discussion we write:

“In this study we found evidence for the retinotopic-like mapping of sound echoes in blind individuals who are experts at perceiving space through sound echoes using clicks, in comparison to blind and sighted controls. Importantly, the degree of retinotopic-like mapping of sound echoes was positively associated with echo localization ability. Those individuals who showed retinotopic-like mapping of sound echoes also showed evidence of spatial mapping of source sounds, even though statistical explanatory power of regression models applied to those data was weaker. This weaker mapping for source sounds, as compared to echo sounds, might be explained by the fact that the typical brain

areas that are responsible for spatially mapping source sounds are still intact in all our participants, and so there is perhaps less need for the brain to undergo neural reorganisation in order to map these in V1. Nonetheless, taken together our data support the conclusion that the characteristic functional topography of a primary sensory area – here, the retinotopic map in primary ‘visual’ cortex – can be used to map sensory input from an atypical modality for a directly analogous task-specific purpose - here, localization of sound. ” Page 14, line 19.

The authors have not mentioned what are the definitions of retinotopy-like maps. Could they explain further what the criteria are to confirm if there is a topographic organization or not.

Our criteria for classifying the organisation as retinotopic-like is that the arrangement of the neural map in primary visual cortex follows a spatial pattern expected based on retinotopic maps of visual space (specifically contralateral mapping and mapping of eccentricity). We operationalized this using the probabilistic atlas of retinotopy by Benson et al (2014), and by correlating the maps we measured with those that are expected based on this atlas. We have now clarified in the introduction what we mean by retinotopic-like:

“We predicted that, for individuals who are blind and experts in echolocation, there will be evidence of retinotopic-like mapping of echo sounds in primary ‘visual’ cortex (i.e. there should be contralateral mapping of stimuli and stimuli at greater eccentricities should be mapped at more anterior points), but not for blind or sighted controls.” Page 4, line 12.

We also include more clarification in the results section:

“We calculated Pearson’s r to quantify the correlation between these predicted eccentricity values and those observed in each of our experimental conditions. We use this method to quantify retinotopic-like organisation because it provides a simple test of whether the pattern of neural mapping is random (i.e. a correlation coefficient of zero) or whether it is comparable to a retinotopic organisation, where higher coefficient values indicate greater retinotopic-like organisation for the mapping of stimuli. All voxels labelled as primary ‘visual’ cortex in the probabilistic atlas were entered into the correlation. These correlation coefficients, along with 95% confidence limits (derived from a bootstrapping method), are shown in Supplementary Table S1. We also confirmed that correlation coefficients expected by chance are indeed zero (results also shown in Supplementary Table S2).” Page 6, line 2.

In response to the reviewer’s comment and also the comments by reviewer 1 we also provide additional analyses in the supplemental material that explore separately the presence of contralateral mapping and eccentricity mapping of echo sounds in the three best-performing expert echolocators to further support our arguments.

MATERIAL & METHODS

The choice of the fMRI acquisition protocol is difficult to understand considering the questions authors aiming to address. The retinotopy maps require continuous stimulation. In the sighted controls visual maps, do not seem to be retinotopically organized, there is not strong contralateral preference as also in the correlation to an atlas shows $r=0.59$. Is sparse-sampling technique advantageous for retinotopy-like mapping? Could the low variance-explained/coherence threshold in the data due to the acquisition protocol?

The reviewer is correct in that a sparse sampling fMRI design is not advantageous for carrying out phase-encoded mapping. Actually, one could consider it not advantageous for any sort of experiment, as the number of acquisitions is always low. The trade-off is between loss of acquired MR samples and the need to avoid over-saturating stimulation in auditory cortices through scanner noise (so that you can actually measure any signal change) and to make auditory stimuli audible. In the case of echolocation, both of these issues apply, but the audibility of echo stimuli is the immediately noticeable limitation that is overcome with the sparse sampling design. We have tested if echolocation stimuli are actually perceptually meaningful in the context of regular (i.e. continuous) MR sampling protocols, and unfortunately they are not. If the scanner continuously images, it is not possible to hear the echoes, and the task becomes totally meaningless. In fact, even the circulatory fan in the scanner impairs how well people can perceive these faint echo sounds, which is why the fan is always switched off in our paradigms (see methods). Simply, we just have no other way to have a perceptual working task in the scanner than using sparse sampling. Because we had no other choice, it was even more important to run all conditions using the same protocol, so as to have all conditions and analyses on equal footing and to determine if the method works (hence the inclusion of the visual condition also).

We have included additional analyses (also in response to suggestion made by reviewer 1) to clearly show that the maps we obtained are statistically reliable. We now show:

- (a) 95% confidence intervals for each of our measured correlation coefficients. These data show that the degree of retinotopic-like mapping of echo sounds in the 3 highest-performing expert echolocators is well above those acquired for the control participants.
- (b) The correlation coefficient that is expected by chance (acquired by randomly shuffling the mapping data and bootstrapping the resulting coefficients) is extremely close to zero with very small confidence interval (around ± 0.02)
- (c) The degree of retinotopic-like mapping for all stimuli types (echoes, source sounds, visual stimuli) measured in V1 positively correlates with that of the same stimuli in V2 (and in V3 for echo and source sounds). This shows that it is very unlikely that the stimulus maps observed in V1 are the results of chance, and are therefore statistically reliable.

We have included these additional analysis and results in the supplemental material and referred to them in the main text as appropriate.

Can the authors be more explicit about why did they choose the correlation coefficient as the measure of behavioral performance? How did they perform the correlation, vector-wise? What does correlation vectorwise mean? Does correlation between the prediction and observe value a good indicator of spatial localization ability? If they had chosen the accuracy as a measure, would they observe the same pattern of results? I suspect the accuracy measure may not be fine-tune, then their experimental design is also a good fit for psychometric curve fitting for instance, and the threshold (extracted from each individuals' curve) could be a good measure for ability of localization. How reliable the correlation analysis, considering the subject pool being very small, $n=5$?

We chose to use the correlation coefficient to quantify psychophysical performance because it is directly analogous to the method that we used to quantify the degree of retinotopic-like mapping of stimuli in primary visual cortex.

Furthermore, whilst a more conventional method to quantify performance, such as proportion correct, is certainly possible, this is not as sensitive as the correlation coefficient because proportion correct would not differentiate small errors (e.g. classifying -40 sound as -20) from large errors (e.g.

classifying -40 sound as +40 sound), whereas the correlation coefficient does take this into account. Given the high variability in performance that we observed using the correlation coefficient as a measure (i.e. from -0.06 to 0.96), we are confident that it is an accurate and sensitive measure of participants' ability to localise these sounds.

Also, it would not be meaningful to fit psychometric functions to our response data. To clarify, we did not obtain proportion correct across variations in the strength of a stimulus property, so we are unable to fit such a function. One possibility would be to fit a function that describes the probability that participants give the rightmost response (i.e. +40) as a function of stimulus position (from -40 to +40), but this would mean that in many cases only a small proportion of the acquired data would be used, discarding all the other information.

For all these reasons, we believe that use of the correlation coefficient to quantify performance in this case is appropriate.

We have added a sentence to the description of the psychophysical task (which has now been moved to the supplemental materials to allow us to include new figures in the main text) to explain the main reason why we chose this measure:

“We chose this correlation measure to quantify psychophysical performance because it is directly analogous to the method that we use to quantify the degree of retinotopic-like mapping of stimuli in primary ‘visual’ cortex. A more conventional method to quantify performance, such as proportion correct, would not be as sensitive as the correlation coefficient, because proportion correct would not differentiate small errors (e.g. classifying -40 sound as -20) from large errors (e.g. classifying -40 sound as +40 sound), whereas the correlation coefficient does take this into account.” Page 18, line 19, supplementary materials.

To quantify the reliability of this measure of performance, we now provide confidence interval data based on bootstrapping. These data are included in supplementary table S5.

95% confidence limits are computed for each correlation. This was done by randomly resampling the data in each correlation 1000 times and calculating correlation coefficients for each resampled dataset, using matlab's *bootstrp* function. Standard error of the mean was then calculated based on these bootstrapped statistics, and this was then used to calculate the upper and lower confidence limits that are reported in the table.

We are not sure what the reviewer means by the questions “How did they perform the correlation, vector-wise? What does correlation vectorwise mean?”

It is not clear to me that why authors did not correlate the obtained sighted control visual retinotopy-like maps with auditory maps. Instead, they relied on Benson atlas as a measure. What are the reasons to use external atlas and then discuss/conclude the way they do in the Discussion (page 10, line 23-24). I suspect they wanted to maintain the same analysis across 3 groups. However, in order to meet their claims in the Discussion, I would suggest showing the same pattern of results when the degree of retinotopy-like maps emerging from correlation between visual and auditory maps.

The reviewer is correct in that we use the external probabilistic atlas in all cases in order to maintain parity in how we quantify retinotopic-like mapping across conditions and participants. It would only be possible to compare observed echo/source maps to observed visual maps in the sighted participants (as we cannot measure them in the blind), and we do not predict (and indeed do not

find) any evidence of retinotopic-like mapping of echo/source sounds in sighted participants. We do not believe, therefore, that this analysis would be informative.

To address the concern raised by the reviewer we have included additional analysis (i.e. 95% CI around correlation values and values expected based on chance; correlation between V1 and V2 and V3 maps) that demonstrate the reliability of our results. These are contained in the supplemental materials and referred to in main text:

“These correlation coefficients, along with 95% confidence limits (derived from a bootstrapping method), are shown in Supplementary Table S1.” Page 6, line 8

“We also confirmed that correlation coefficients expected by chance are indeed zero (results also shown in Supplementary Table S2).” Page 6, line 9

“Whilst for theoretical reasons the main focus of our manuscript is on V1, we also carried out the same mapping analyses in the second and third visual cortex (V2 and V3), each of which contain a retinotopically organised map of visual space [28] and are directly connected to auditory cortex [29-31]. We also measured similarity of maps in V2 and V3 to those observed in V1. For those EEs where we found retinotopic-like mapping of echo and source sounds in primary ‘visual’ cortex, we also find it in V2 and V3, suggesting that retinotopic-like mapping of echo and source sounds extends beyond the primary ‘visual’ area. Furthermore, high similarity between maps in V1 and V2/V3 strongly suggest that maps in primary ‘visual’ cortex are reliable and not the result of low statistical power. All of this is reported in the Supplementary Materials.” Page 14, line 8.

Mapping data for V2 and V3 are shown in supplementary tables S3 and S4, respectively.

Related to the low coherence threshold, the authors have mentioned the upper limit, due to the visual retinotopy-like mapping. In order to distinguish the observed retinotopy-like mapping in the echolocators for echo- sounds, it would be crucial to have lower limit for each individual’s maps. There are no whole-brain results, so it is hard to understand how much of the results are noise or task-related. One way to test this, is to show whole brain results, and probably taking region-of-no-interest coherence. The coherence in that region could be a threshold/baseline. Then assigning this value as a “baseline” and/or using to perform stats to understand each individual’s’ variance explain/coherence is significantly different than the one observed in no-interest region (noise). Another way would be splitting the data into two halves and running correlation to show that the observation is different from noise.

To address the reviewer’s concern we have provided additional analyses and results (i.e. 95% CI around correlation values; correlation values expected based on chance; correlation between V1 and V2/V3 maps) that demonstrate the reliability of our results, and that address the concerns raised here, and those raised by reviewer 1 .

Taken together our results suggest that the observed neural maps in primary visual cortex are reliable and not the result simply of low statistical power.

The cross-correlation results are within the mask of V1, V2, V3? Is there a measure for that, e.g. FreeSurfer based? I’m guessing it’s from Benson atlas, they masked their functional results with. Nevertheless, it’s not clear in the manuscript. A way would be representing in the Figure 1 the landmark of Benson atlas.

That is correct – we carry out the phase-encoded mapping only for voxels that are within the mask of V1. The Benson et al (2014) atlas that we use to define this mask predicts the anatomical location of the stria of Gennari (the anatomical marker of V1) with reference to cortical surface topology (Hinds et al, 2008). Benson et al provide freely available software for applying this mask to a subject's data that has been processed using FreeSurfer's recon-all pipeline (https://hub.docker.com/r/nben/occipital_atlas/)

Ref:

Hinds OP, Rajendran N, Polimeni J, Augustinack J, Wiggins G, Wald L, Diana Rosas H, Potthast A, Schwartz EI, Fischl B. Accurate prediction of V1 location from cortical folds in a surface coordinate system. *NeuroImage*. 2008; 39(4):1585–1599

We have added a statement to clarify this in the manuscript:

“The atlas identifies the estimated anatomical location of the stria of Gennari (anatomical marker of primary ‘visual’ cortex) with reference to cortical surface topology [24]. It then assigns a numeric value to each voxel in this area to represent the location in space (in visual degrees) that the voxel is most likely to represent, ranging from -90 (most peripheral left space) to +90 (most peripheral right space).” Page 5, line 23.

We have now included on each image of the brain an outline showing the perimeter of V1, as defined by the atlas that we used to mask the functional data.

It would help the reader to understand the perceptual difference between auditory echo localization and sound source localization if the stimuli were also provided.

We provide waveform plots (Figure 5) that illustrate the difference across conditions and we also describe how stimuli were made for each participant (i.e. set up and procedure for binaural recordings; sound editing etc).

We have also amended captions for Figure 4 and 5 to improve clarity.

We have also added additional acoustic analysis for the sound stimuli in the supplemental materials.

We are happy to share stimuli with others upon reasonable request and have included a data sharing statement to that effect.

For the sake of replicability of the analysis, and clarity of the technique that was used, it is essential to have open access to the custom Matlab/R scripts. E.g. which function do they use for multiple linear regression analysis, are assumptions for linear regression analysis met? The correlation values are normally distributed? Please see the simple R script in the link as an example.

https://www.dropbox.com/s/7hsctigt2rbgd82/LM_assumption_checks.R?dl=0

We now include custom matlab functions that were used in the fMRI analysis. These are codes that are designed to be used with the open source software suite ‘mrTools’ (Gardner Lab, Stanford University, USA). The functions that are included are as follows:

corAnal – designed to replace the function corAnal in the mrTools suite. This function includes a call to the following function:

computeCoranal –designed to replace the function computeCoranal in the mrTools suite. This function includes a call to the following function:

myXcorr – this is an entirely custom built function that carries out phase-encoded mapping on sparse sampling mri data in a way that is compatible with the mrTools analysis pipeline.

The multiple regression was carried out using SPSS (v22). Analyse -> Regression -> Linear, with settings set to default. The dataset used in this regression is included in the submission.

The assumptions of using multiple linear regression are met in our case:

First, the relationship between behavioural performance and retinotopic-like mapping (for expert echolocators) is linear, as can be seen by the scatter plots shown in the main manuscript.

Second, inspection of the Q-Q plots for the residuals shows a normal distribution.

Regression for retinotopic-like mapping of echoes:

Regression for retinotopic-like mapping of source sounds:

Third, although there is multicollinearity between the interaction term and its component variables (VIF value > 10), this is to be expected because this term is a product of those component variables. This is not an issue for our interpretation of the effect of the interaction term, however, because centering the variables first (de-meaning them) before computing the interaction term eliminates this collinearity, but the effect of the interaction is the same as when the factors are not centered. See Robinson and Schumacker (2009)

Ref: Robinson, C., and R. E. Schumacker. 2009. Interaction effects: Centering, variance inflation factor, and interpretation issues. *Multiple Linear Regression Viewpoints* 35 (1):6–11

Fourth, scatterplots of the residuals show that there is a reasonable assumption of homoscedasticity in the data, (i.e. no relationship between the predicted values (x-axis) and the standardized residuals (y axis)):

Regression for retinotopic-like mapping of echoes:

Regression for retinotopic-like mapping of source sounds:

Can they explain briefly why they chose for multiple regression analysis and specifically that model?

We have now included a statement in the results as to why we chose the method of multiple linear regression:

“We used the method of multiple linear regression here because it allows us to include all relevant factors in a single analysis.” Page 11, line 15.

When multiple regression analyses results are reported the following items were missing: effect sizes, confidence intervals, etc... Inserting variance explained value right after the model parameters/regressors definitions, would potentially avoid the confusion. Is the variance explained here R^2 ?

In addition to our previous analysis which reported standardized coefficients as measures of effect size, p-values and test-statistics, and model fits, we now also include unstandardized B values and 95% confidence intervals for the effects reported in the multiple linear regression.

Yes, the variance explained here is R^2 . We hope that this is now clear in the manuscript: e.g. *“The overall model explained 80.9% of the variance (R^2)”* Page 11, line 23.

RESULTS

How do authors explain the sound sources evoked less response/location preference compared to the echo sounds (i.e. SC2, SC3, SC4)?

Again thank you for pointing out that we needed to explain more.

Please see our response to your very first major concern which addresses this issue.

I'm guessing that stimuli energy levels across two auditory tasks are not equalized to be ecologically valid. Do the auditory stimuli of the 8 spatial locations show the same frequency? Would frequency or energy difference across stimuli explain their results?

Overall intensity of stimuli was matched across conditions, i.e. as stated in manuscript (now in the supplemental materials because we needed to be mindful of manuscript length to meet the page limit):

“Sounds were played to participants at a level at which the highest peak intensity was presented at 80 dB SPL, and the same sound level was used for echo and source-sound conditions, and also for psychophysical tests outside the scanner.” Page 1, line 24, supplementary materials

Stimuli were binaural. Due to the nature of the stimuli (see also waveform plots) binaural intensity differences are more pronounced for source (6.8 dB; SD 4.2) as compared to echo sounds (1.3dB; SD: 1.2), because in echo sounds the click is always central as it is emitted from the location of the mouth, and the echo is comparably weaker compared to the source sound. Please also see waveform plots in manuscript. This cannot be avoided due to the nature of the echo-acoustic process. To address the reviewer's concern we have put this information into the supplemental materials:

“Due to the nature of the stimuli, the average binaural intensity difference across stimulus positions was greater for source sounds (6.8 dB; SD 4.2) compared to echo sounds (1.3 dB; SD: 1.2). This is because, in the echo sounds the click is always central as it is emitted from the location of the mouth, and the echo is weaker compared to the source sound from the same position.”

...

Importantly, however, since any acoustic differences in the stimuli apply to all participants in our study, they cannot explain the association between any neural mapping we observed and performance in the echolocation task.” Page 22, line 3, supplementary materials

Further, to address the reviewer’s concern we performed a spectral analysis of the recorded click and source sounds, as well as a comparison of the echo (averaged across the two channels) across four positions of the object (+05, +10, +20, +40). Compared to the clicks in the echo recordings, those in the source sound recordings contain more energy in lower frequency bands. This is to be expected given the directional characteristics of the speaker, which was positioned to face away from the participant’s ears during recording of the echo sounds, but was positioned to face towards the participant’s ears during recording of the source sounds. There are minimal changes in the spectrum of the echoes across different object positions (small differences are expected as more high frequency energy will be returned from objects that are positioned more centrally, again expected due to directional characteristics of the speaker).

These data are in the supplemental material and referred to in the main text:

“Acoustic analysis of these sound stimuli, along with details on equipment used for recording and playback, are included in the Supplementary Materials.” Page 19, line 19.

Importantly, since these effects apply to all participants in our study, they cannot explain the association between any neural mapping we observed and performance in the echolocation task.

The click emission is not broadband stimulus, might explain the weak activity in sighted and blind participants?

The click in fact is a broadband and complex acoustic stimulus. We have now included spectral analysis of recordings of the clicks in the supplemental materials (as mentioned above) to illustrate this.

Also, the same click stimulus was used in the source sounds, and all participants performed very well in the source localization task on the behavioural level. This rules out that they simply could not hear where click sounds came from.

What are the differences between stimuli (echo vs. source sounds)?

We would like to refer the reviewer to the description of the difference in how the echo/source sounds were recorded in the text:

“For each set of echo sound recordings, a sound reflector (wooden disk, 17.5 cm diameter) was placed directly facing the participant at ear height (mounted on a pole, 1 cm diameter) at a radial distance of 1 m and at one of 8 angular (azimuth) positions - L40°, L20°, L10°, L05°, R05°, R10°, R20°, and R40°, where L and R correspond to left and right relative to the participant’s orientation. In contrast, for each set of source sound recordings, the loudspeaker was positioned at one of the 8 angular positions, directly facing the participant. The same artificial click emission used in the echo

sound recordings was used for the source sound recordings, but here it is emitted directly towards the participant and with no other sound reflectors present.” Page 19, line 4.

We would also like to refer the reviewer to the graph in figure 5 which illustrates the difference in acoustic content between the echo and source sound recordings.

We have also amended captions for Figure 4 and 5 to improve clarity.

Please also see our additional spectral and intensity analyses now included in the supplementary materials.

Were authors expecting ipsilateral response in the visual retinotopy experiment in sighted controls? Can they elaborate what could it mean? Did the results show contralaterality in the visual cortex? Maybe the authors had performed such analysis, but it is not clear in the manuscript. Such analyses would increase the reliability of the results, since with visual inspection some sighted controls seems to not follow “typical” retinotopic organization.

We were not expecting ipsilateral mapping of visual stimuli in primary visual cortex. We were, however, expecting there to be a degree of noise in the data due to the limitations of the sparse sampling design. Nonetheless, the overall degree of retinotopic-like mapping of these visual stimuli is good ($r=0.59$) and our additional analyses (i.e. 95% CI, comparing obtained visual mapping to mapping expected based on chance and correlation between maps in V1 and V2/V3) demonstrate reliability.

The degree of sound source retinotopy-like map was correlated with echo localization ability instead investigating correlations with sound localization ability. What was the hypothesis for this analysis?

We use Echo Localization Ability as a predictor here, not Source Localization Ability, because our hypothesis concerns whether echolocation ability predicts the degree of mapping of echo and/or source sounds. Furthermore, there is very little variation across participants’ ability to localize source sounds (they all do very well at the task), making it a poor predictor.

For clarification, we have added the following statement as a footnote in the manuscript:

“We use Echo Localization Ability as a predictor here, not Source Localization Ability, because our hypothesis concerns whether echolocation ability predicts the degree of mapping of echo and/or source sounds.” Page 13, footnote number 1.

Figures: in such inflated brain, the curvatures are not visible, please consider smoothing less.

We would prefer not to include curvature on these images, as doing so can partially obscure some of the maps. Nonetheless, we have now included an outline that marks the boundary of V1 (as measured by the Benson et al atlas), which will hopefully help the reader to interpret the neural anatomy shown.

Also it would be better for the reader to see Figure 1 and 2 together to compare the maps visually.

Figure 1 and figure 2 are now combined into a single figure.

Furthermore, without visual inspection of the sighted controls or blind individuals' maps, I find the results are not informative in the main text. I would highly recommend adding at least one of the two groups' results into main manuscript.

Figures for blind controls are now included in the main manuscript

By visual inspection, EE3 and EE4 are the two participants showing not consistent maps across echo sounds and source sounds, especially in the right hemisphere, while three other subjects showing consistent/similar maps for both echo and source sounds. Is there any hypothesis for such results?

We are only able to speculate as to the reasons that this might be the case. One possible reason is that the difference in EE4 at least can be explained by relatively poor echo localisation performance (compared to the other EEs). As for EE3, it might be related to the slightly later age at which they developed their echolocation ability.

As for the more pronounced difference in right V1, it is perhaps worth noting that previous research has shown that the right hemisphere of V1, in comparison to the left, shows greater activity in response to echo sounds compared to control sounds not containing echoes (e.g. Thaler et al, 2011), but since sample sizes are low any laterality interpretation is speculative.

DISCUSSION

Considering echolocator experts show retinotopy-like mapping for echo sounds, what is the explanation or theoretical framework of weaker sound-source evoked retinotopy-like maps in the EVC of echolocators?

This also relates to the first major comment by the reviewer.

We would first like to clarify that our neural data are the result of a cross correlation analysis, and as such they do not represent the strength of activity in response to auditory stimulation per se, but rather how consistently a specific voxel responded to a specific stimulus position. Thus, these data should not be taken as indicators of the strength of activity in primary visual cortex to sound, as you could for example measure in an experimental design that would compare responses to acoustic stimulation to responses in a non-acoustic control condition.

We have added to the results to clarify this:

"It is important to note that neural data are the result of a cross correlation analysis, and as such they do not represent the strength of activity in response to auditory stimulation per se, but rather how consistently a specific voxel responded to a specific stimulus position." Page 5, line 8.

Nonetheless, it is important for us to clarify our interpretation why we observed more consistent mapping for echo sounds as compared to source sounds.

As outlined in the introduction, we used echolocation as perceptual paradigm alongside source localization because in this way the inclusion of blind echolocators as well as blind non-echolocators enabled us to investigate the effects of blindness and expertise simultaneously (in this case: expertise in echolocation). Any retinotopic-like mapping in V1 in any condition is in principle consistent with task specificity in V1 (rather than modality specificity).

Our results show that mapping of echo sounds in V1 is associated with echo expertise. They also show that mapping of source sounds in V1 is associated with echo expertise, though to a lesser degree.

The finding that in those individuals who did show mapping, we observed weaker mapping for source sounds, as compared to echo sounds, might be explained by the fact that the typical brain areas that are responsible for spatially mapping source sounds are still intact in all our participants, and so there is perhaps less need for the brain to undergo neural reorganisation in order to map these in V1.

To make this very clear in our discussion we write:

“In this study we found evidence for the retinotopic-like mapping of sound echoes in blind individuals who are experts at perceiving space through sound echoes using clicks, in comparison to blind and sighted controls. Importantly, the degree of retinotopic-like mapping of sound echoes was positively associated with echo localization ability. Those individuals who showed retinotopic-like mapping of sound echoes also showed evidence of spatial mapping of source sounds, even though statistical explanatory power of regression models applied to those data was weaker. This weaker mapping for source sounds, as compared to echo sounds, might be explained by the fact that the typical brain areas that are responsible for spatially mapping source sounds are still intact in all our participants, and so there is perhaps less need for the brain to undergo neural reorganisation in order to map these in V1. Nonetheless, taken together our data support the conclusion that the characteristic functional topography of a primary sensory area – here, the retinotopic map in primary ‘visual’ cortex – can be used to map sensory input from an atypical modality for a directly analogous task-specific purpose - here, localization of sound.” Page 14, line 19.

Could the authors discuss briefly what are possible explanations of the observed differences in echo localization vs. sound source localization in the brain?

We thank the reviewer for pointing out the need for additional explanations. We have added to our manuscript text to address this:

“This weaker mapping for source sounds, as compared to echo sounds, might be explained by the fact that the typical brain areas that are responsible for spatially mapping source sounds are still intact in all our participants, and so there is perhaps less need for the brain to undergo neural reorganisation in order to map these in V1.” Page 14, line 24.

Please see also our response to the previous comment and major comment 1.

Page 10, line 21-24, the authors claimed that retinotopic map in the EVC can be used to map sensory nonvisual input. However, there is no analysis to show that retinotopy maps in the visually deprived individuals stem from the typically retinotopic organization, e.g. assuming the maps are located only in V1, are the zones in the visual maps showing contralateral response showing same/similar preference in the auditory maps?

The reviewer is correct that we cannot irrefutably conclude that the retinotopic-like mapping in V1 is due to the same retinotopically organised map that maps visual space in the sighted brain, as

opposed to an alternative view in which a novel map with comparable retinotopic-like organisation spontaneously originates in primary visual cortex of the expert echolocators.

Given that the mapping we observe is similar to probabilistic mapping of retinotopy expected based on anatomy, however, and given that intrinsic retinotopic connectivity in early visual cortex exists even in the congenitally blind brain (Striem-Amit *et al*, 2015), we believe that the most parsimonious explanation is that retinotopic topography is used here to map sound echoes.

To clarify our reasoning we write in the discussion:

“What mechanisms might underlie the spatial mapping of sounds in primary ‘visual’ cortex? We propose that, because the spatial pattern of this mapping is directly comparable to that of visual space in the sighted brain (i.e. it follows a retinotopic-like pattern), it is likely that this mapping takes advantage of the intrinsic retinotopic organisation of early visual cortex, which forms even in the complete absence of visual input [17]. An analogous neural map of space does not exist in primary auditory cortex (it is tonotopic [40]), and so the map of space in primary ‘visual’ cortex might be the most suitable cortical site on which to map spatial location as conveyed through sound echoes. One explanation for cross-modal reorganization of ‘visual’ cortex is that primary ‘visual’ cortex is more strongly connected to other sensory areas, including auditory cortex [41], in cases of early or prolonged vision loss. These greater structural connections might confer additional functional connectivity between auditory and visual areas when an individual performs a relevant auditory task, but not necessarily at rest [42].” Page 16, line 12.

The inter-subject variability seems to be high, even in auditory tasks only, same zones seem to not show same preference of space in the blind individuals (see Material method section, point 4, suggestion for performing correlation analysis only in the sighted group to investigate whether same voxels showing XX eccentricity or contra-laterality preference in the visual maps, would show similar preference in auditory locations?).

The reviewer is correct that there is variability in the data.

As we mentioned in our response to their previous comment that this one refers to, we do not believe that this analysis would be informative. Specifically, because we do not predict (and indeed do not find) any evidence of retinotopic-like mapping of echo/source sounds in sighted participants, then performing a statistical comparison of these maps to those obtained for visual stimuli in the same participant group would not produce an informative result.

The additional analyses that we have provided in the revised version of this manuscript (i.e. 95% CI around correlation values and values expected based on chance; correlation between V1 and V2/V3 maps) address the reviewer’s concerns, however, as well as concerns raised by reviewer 1, and support our conclusions.

The results also showed little evidence to retinotopy-like mapping in blind individuals both for echo and source sounds. Considering the importance of spatial perception in blind individuals, it seems rather surprising that early blind participants did not show retinotopy-like maps in the visual cortex. What could be the explanation? Could it be due to low number of participants? And/or the fMRI acquisition protocol? Considering the previous studies in the field focused on the early blind individuals, and how non-visual input functionally mapped on the visual cortex, how do the authors explain their results (Page 11, line 9-20). Page 12, line 4-5, 16-17, could the authors further discuss the differences in the processing of spatial localization in echo-locators and non-echolocator blind individuals?

Thank you for pointing out that additional explanation is needed. Please see also our response to your previous related comments.

We would first like to clarify that our neural data are the result of a cross correlation analysis, and as such they do not represent the strength of activity in response to auditory stimulation *per se*, but rather how consistently a specific voxel responded to a specific stimulus position. Thus, these data should not be taken as indicators of the strength of activity in primary visual cortex to sound, as you could for example measure in an experimental design that would compare responses to acoustic stimulation to responses in a non-acoustic control condition.

We have added a section in the results to clarify this:

“It is important to note that neural data are the result of a cross correlation analysis, and as such they do not represent the strength of activity in response to auditory stimulation per se, but rather how consistently a specific voxel responded to a specific stimulus position.” Page 5, line 8.

As outlined in the introduction, we used echolocation as perceptual paradigm alongside source localization because in this way the inclusion of blind echolocators as well as blind non-echolocators enabled us to investigate the effects of blindness and expertise simultaneously (in this case: expertise in echolocation). Any retinotopic-like mapping in V1 in any condition is in principle consistent with task specificity in V1 (rather than modality specificity).

Our results show that mapping of echo sounds in V1 is associated with echo expertise. They also show that mapping of source sounds in V1 is associated with echo expertise, though to a lesser degree (i.e. less explained variance). Whilst this clearly demonstrates task specificity (rather than modality specificity) in V1, it also suggests that this task specificity does not arise by default in response to blindness (as in that case every blind participant should have shown mapping for sound sources as everyone can perceive them well), but that for this task-specific adaptation of V1 proficiency in processing sensory input (here: echolocation expertise) might be the critical factor. As we state in our discussion, this finding and interpretation is in line with previous studies showing cross-modal activity in early ‘visual’ cortex from proficient sensory skill use (e.g. Braille, or sensory substitution devices; monaural sound localization).

Also in response to reviewer 1 comment (which we think is related) we now write in the discussion:

“It is clear from the results of the present study, however, that blindness is not sufficient to explain the cross-modal recruitment of primary ‘visual’ cortex, as blind control participants did not show evidence of retinotopic-like mapping either for source or echo sounds. It is important to note, however, that there was variation in the onset of blindness in our sample, i.e. BC5 is classified as late blind and EE3, BC2 and BC4 (whilst early blind) have no clear onset of total blindness, and that also the age at which EEs started using echolocation is variable. However, there was no relationship between degree of retinotopic-like mapping and age at onset of blindness in either group, and in our EE sample there was also no relationship between neural mapping and age at onset of echolocation use. Furthermore, because the degree of retinotopic-like mapping of sound in the expert echolocators was positively associated with echo localisation ability, proficiency in processing sensory input might be the critical factor in the neural reorganisation of primary sensory areas. This is in line with previous studies showing cross-modal activity in early ‘visual’ cortex from proficient sensory skill use (e.g. Braille, or sensory substitution devices [10]; monaural sound localization [12]).” Page 16, line 23.

The finding that in those individuals who did show mapping, we observed weaker mapping for source sounds, as compared to echo sounds, might be explained by the fact that the typical brain areas that are responsible for spatially mapping source sounds are still intact in all our participants, and so there is perhaps less need for the brain to undergo neural reorganisation in order to map these in V1.

To make this very clear in our discussion we write:

“In this study we found evidence for the retinotopic-like mapping of sound echoes in blind individuals who are experts at perceiving space through sound echoes using clicks, in comparison to blind and sighted controls. Importantly, the degree of retinotopic-like mapping of sound echoes was positively associated with echo localization ability. Those individuals who showed retinotopic-like mapping of sound echoes also showed evidence of spatial mapping of source sounds, even though statistical explanatory power of regression models applied to those data was weaker. This weaker mapping for source sounds, as compared to echo sounds, might be explained by the fact that the typical brain areas that are responsible for spatially mapping source sounds are still intact in all our participants, and so there is perhaps less need for the brain to undergo neural reorganisation in order to map these in V1. Nonetheless, taken together our data support the conclusion that the characteristic functional topography of a primary sensory area – here, the retinotopic map in primary ‘visual’ cortex – can be used to map sensory input from an atypical modality for a directly analogous task-specific purpose - here, localization of sound.” Page 14, line 19.

MINOR

Introduction

Page 3, line 8-10, 17-19 need more clarity, clear transition and/or further explanation.

Based on the reviewer’s useful feedback we have now reworded these sections in order to improve clarity:

“Specifically, whilst it has been shown that primary sensory areas may process input from atypical modalities [6 – 13], it remains to be shown that these areas can use this input in order to perform a specific task that would ordinarily be performed with input from its typical modality.” Page 3, line 7.

“There is evidence based on patterns of non-stimulus driven activity (i.e. resting state data) that the intrinsic retinotopic organisation of early visual cortex can be preserved even in congenitally blind individuals [17]. This means that neurons within early visual cortex are connected in a way that is consistent with a retinotopic organisation. It is unknown, however, whether this intrinsic connectivity can be adapted by a non-visual modality for the mapping of space.” Page 3, line 15.

Results

Page 5, line 16-23 needs more clarification.

Based on the reviewer's useful feedback we have now reworded these sections in order to improve clarity:

"The atlas has a very low prediction error for eccentricity (0.51° of visual angle), in that it accurately predicts the location and layout of a retinotopic map that is acquired through functional measurement [24]. The atlas identifies the estimated anatomical location of the stria of Gennari (anatomical marker of primary 'visual' cortex) with reference to cortical surface topology [24]. It then assigns a numeric value to each voxel in this area to represent the location in space (in visual degrees) that the voxel is most likely to represent, ranging from -90 (most peripheral left space) to +90 (most peripheral right space). We calculated Pearson's r to quantify the correlation between these predicted eccentricity values and those observed in each of our experimental conditions." Page 5, line 21.

Discussion

Page 12, line 4-5, can authors cite more studies to support their prediction? According to their prediction/claim, one could also predict that early blinds could show map-like organization for spatial perception in EVC.

Please see also our response to previous comments regarding our results, i.e. that mapping is associated with performance in EEs, and weaker mapping for sources than echo sounds, i.e. that we do not think that blindness alone is sufficient for such reorganization to occur, but that proficiency in a sensory skill is relevant and that the typical brain areas that are responsible for spatially mapping source sounds are still intact in all our participants, and so there is perhaps less need for the brain to undergo neural reorganisation in order to map these in V1.

In addition to the revisions addressing the above, and outlined in response to those previous comments, based on this specific comment and useful feedback we have also reworded this specific statement:

"Given that previous studies have shown that functionally reorganised visual areas are in fact causally involved in processing non-visual information – e.g. through the application of neurostimulation [38 – 39] – it is possible that primary 'visual' cortex is functionally necessary for the perception of space through sound, and in particular sound echoes, in some expert echolocators." Page 16, line 8.